# Nicotine-mediated OTUD3 downregulation inhibits VEGF-C mRNA decay to promote lymphatic metastasis of human esophageal cancer

Meng Wang[1,4], Yue Li[1,4], Yunyun Xiao[1,4], Muwen Yang[1,4], Jinxin Chen[1], Yunting Jian[1], Xin Chen[2], Dongni Shi[1], Xiangfu Chen[1], Ying Ouyang[1], Lingzhi Kong[1], Xinjian Huang[1], Jiewen Bai[1], Chuyong Lin ⬤ [1,3 ✉] & Libing Song ⬤ [1,2 ✉]

Nicotine addiction and the occurrence of lymph node spread are two major significant factors associated with esophageal cancer's poor prognosis; however, nicotine's role in inducing lymphatic metastasis of esophageal cancer remains unclear. Here we show that OTU domain-containing protein 3 (OTUD3) is downregulated by nicotine and correlates with poor prognosis in heavy-smoking esophageal cancer patients. OTUD3 directly interacts with ZFP36 ring finger protein (ZFP36) and stabilizes it by inhibiting FBXW7-mediated K48-linked polyubiquitination. ZFP36 binds with the VEGF-C 3-'UTR and recruits the RNA degrading complex to induce its rapid mRNA decay. Downregulation of OTUD3 and ZFP36 is essential for nicotine-induced VEGF-C production and lymphatic metastasis in esophageal cancer. This study establishes that the OTUD3/ZFP36/VEGF-C axis plays a vital role in nicotine addiction-induced lymphatic metastasis, suggesting that OTUD3 may serve as a prognostic marker, and induction of the VEGF-C mRNA decay might be a potential therapeutic strategy against human esophageal cancer.

[1] Department of Experimental Research, State Key Laboratory of Oncology in South China, Collaborative Innovation Center for Cancer Medicine, Sun Yat-sen University Cancer Center, 510060 Guangzhou, China. [2] Key Laboratory of Protein Modification and Degradation, School of Basic Medical Sciences; Guangzhou Institute of Oncology, Tumor Hospital, Guangzhou Medical University, 511436 Guangzhou, China. [3] Guangdong Esophageal Cancer Institute, 510060 Guangzhou, China. [4] These authors contributed equally: Meng Wang, Yue Li, Yunyun Xiao, Muwen Yang. ✉email: linchy@sysucc.org.cn; songlb@sysucc.org.cn

Esophageal cancer ranks seventh in annual new incidence and sixth in mortality globally, with half of which occurs in China and a poor 5-year survival rarely >20%[1]. Epidemiologic studies indicate that cigarette smoking is a significant prognostic factor, suggesting that smoking may further contribute to esophageal cancer's malignant progression[2–4]. On the other hand, the presence of tumor cells in regional or sentinel lymph nodes (LNs) is the most crucial feature of malignant esophageal cancer, directly inducing therapy resistance and tumor relapse[5,6]. Notably, retrospective studies showed that the smoking history is associated with LN metastasis status at the time of diagnosis and post-therapy in patients with esophageal[7–10], oral[11], lung[12], and cutaneous[13] cancers. Moreover, experimental data revealed that nicotine exposure promoted LN metastasis in head and neck squamous cell carcinoma[14,15]. However, the role of nicotine in inducing lymphatic metastasis of esophageal cancer remains unclear. Further research focusing on the mechanisms and key regulators in this process is necessary to design therapeutic strategies with the potential to control esophageal cancer progression.

Accumulating evidence indicates that cancer cells can enhance the development of lymphatic vessels within and near tumors, known as tumor-induced lymphangiogenesis, thereby facilitating the LN dissemination of tumor cells[16–18]. Tumors provoke the growth and remodeling of lymphatic vessels by secreting pro-lymphangiogenic growth factors, such as the vascular endothelial growth factor C (VEGF-C). VEGF-C binds and activates the VEGFR3 signaling, promoting the proliferation, migration, and vascular permeability of lymphatic endothelial cells (LECs)[16,19]. Consistently, VEGF-C is upregulated and promotes lymphatic metastasis in various human cancers, including esophageal[20], bladder[21], breast[17], and colorectal[22]. For example, we previously reported that transducin (β)-like 1 X-linked receptor 1 (TBL1XR1) increased VEGF-C transcription to promote lymphatic metastasis of esophageal cancer[20]. Importantly, the VEGF-C blockage substantially reduced LN metastasis in mouse xenograft models[20,21], suggesting that targeting VEGF-C may be a promising strategy. Unveiling the molecular mechanism underlying VEGF-C expression will hopefully open new avenues to prevent or overcome esophageal cancer.

Notably, recent studies reveal that decay of messenger RNA (mRNA) is emerging as a critical step for the expression of genes encoding cytokines[23]. Most of these transcripts are short-lived and contain conserved AU-rich elements (AREs) in their 3'-untranslated region (UTR). The AREs function as the cis-acting mediators and are recognized by trans-acting RNA-binding proteins (RBPs) such as ZFP36 ring finger protein (ZFP36/TTP)[23–25]. ZFP36 uses its tandem CCCH zinc fingers to bind the AREs of substrate mRNAs. ZFP36 further recruits the mRNA decapping factor DCP2[26], CCR4-NOT deadenylase complex[27], and RNA-degrading exosome[28], thus mediating the rapid turnover of client mRNAs. For instance, ZFP36 has been identified as a direct suppressor of various cytokine mRNAs, including VEGF-A, interleukin (IL)-6, IL-8, cyclooxygenase-2, interferon-γ, and tumor necrosis factor-α[23]. Notably, previous studies demonstrated that ZFP36 played tumor-suppressive roles in animal models and human cancers[29–32]. In particular, ZFP36 directly induced mRNA decay of VEGF-A to repress angiogenesis (blood vessels)[33]; however, its role in lymphangiogenesis (lymphatic vessels) remains unclear. Intriguingly, a negative correlation between ZFP36 and VEGF-C was observed in uremic rats with peritoneal dialysis[34]. Although VEGF-C transcription has been well studied, whether its mRNA decay is regulated by ZFP36 to acquire high tumor-induced lymphangiogenesis capacity remains obscure.

Deubiquitinases (DUBs) can exert pleiotropic functions by trimming or removing ubiquitin chains from diverse substrate proteins[35]. Approximately 100 human DUBs are known so far, some of which are promising therapeutic targets[35]. Interestingly,

we recently identified that OTU domain-containing protein 3 (OTUD3) was the most significantly changed DUB gene induced by the cigarette smoke extract (CSE). OTUD3 can deubiquitinate K48-linked ubiquitination to stabilize proteins[36–38] and K63- or K11-linked ubiquitination to regulate protein activity[39,40]. The malfunction of OTUD3 facilitates the development and progression of human diseases, including cancers[36–38]. Notably, OTUD3 was downregulated in human cancers, contributing to enhanced cell proliferation, anti-apoptosis, and invasion[36,37]. However, the role of OTUD3 in smoking-mediated esophageal cancer progression remains unclear.

Here we show that nicotine-induced OTUD3 downregulation correlates with lymphangiogenesis, LN metastasis, and poor prognosis in smoking esophageal cancer patients. OTUD3 inhibits the K48-linked ubiquitination and stabilizes ZFP36, which facilitates VEGF-C mRNA decay. Thus, ectopic expression of OTUD3 or ZFP36 abrogates nicotine-induced VEGF-C expression and lymphatic metastasis. These findings uncover a mechanism for nicotine addiction in facilitating lymphatic metastasis of esophageal cancer and suggest that induction of VEGF-C mRNA decay is a potential therapeutic strategy.

## Results

**OTUD3 downregulation by nicotine correlates with LN metastasis.** DUBs play critical roles in developing and progressing human cancers and might be targetable vulnerabilities[35]. Interestingly, our recent RNA-seq analysis showed that OTUD3 was the most significantly regulated DUB gene ($\log_2 Fc > 1$, $P < 0.05$) by CSE in esophageal cancer cells (Fig. 1a). Consistently, CSE treatment downregulated the OTUD3 mRNA and protein expression in multiple esophageal cancer cell lines but not in the normal esophageal epithelial cell (NEEC) (Fig. 1b). Notably, the main smoke substance nicotine and its derivatives dramatically reduced OTUD3 expression, while other tobacco carcinogens, such as acetaldehyde and 4-aminobiphenyl, had no significant effects on OTUD3 expression (Fig. 1c). These findings reveal that nicotine could specifically downregulate OTUD3 in esophageal cancer cells.

Nicotine and its derivatives bind to the nicotinic acetylcholine receptors (nAChRs) to activate downstream signaling pathways including phosphoinositide-3 kinase (PI3K)/AKT, mitogen-activated protein kinase, and Janus-activated kinase/signal transducer and activator of transcription factor 3 (JAK2/STAT3)[41]. Interestingly, the PI3K inhibitor LY294002, but not the BRAF inhibitor or JAK2 inhibitor, abrogated nicotine-mediated OTUD3 downregulation (Supplementary Fig. 1a, b). Indeed, the putative promoter of OTUD3 contained four potential binding sites of FOXO1 (Supplementary Fig. 1c), suggesting that nicotine regulated OTUD3 expression through PI3K/AKT/FOXO1 signaling. As expected, promoter-reporter assays indicated that FOXO1 promoted OTUD3 transcription mainly via the third binding site (Supplementary Fig. 1d). Consistently, nicotine reduced the OTUD3 promoter activity and decreased the enrichment of FOXO1, p300, H3K27ac, and RNA polymerase II (pol II) on the OTUD3 promoter (Supplementary Fig. 1e, f).

The alpha 7 nAChR is one of the most important nAChRs and is associated with human cancers[42]. Notably, we found the alpha7 nAChR was upregulated in esophageal cancer cells (Supplementary Fig. 1g). Moreover, the alpha7 nAChR antagonist α-Bungarotoxin substantially impaired nicotine-induced Akt activation, FOXO1 inhibition, and OTUD3 downregulation in esophageal cancer cells (Supplementary Fig. 1h, i). These findings indicate that the alpha7 nAChR is essential for nicotine to regulate the PI3K/AKT/FOXO1/OTUD3 axis in esophageal cancer cells.

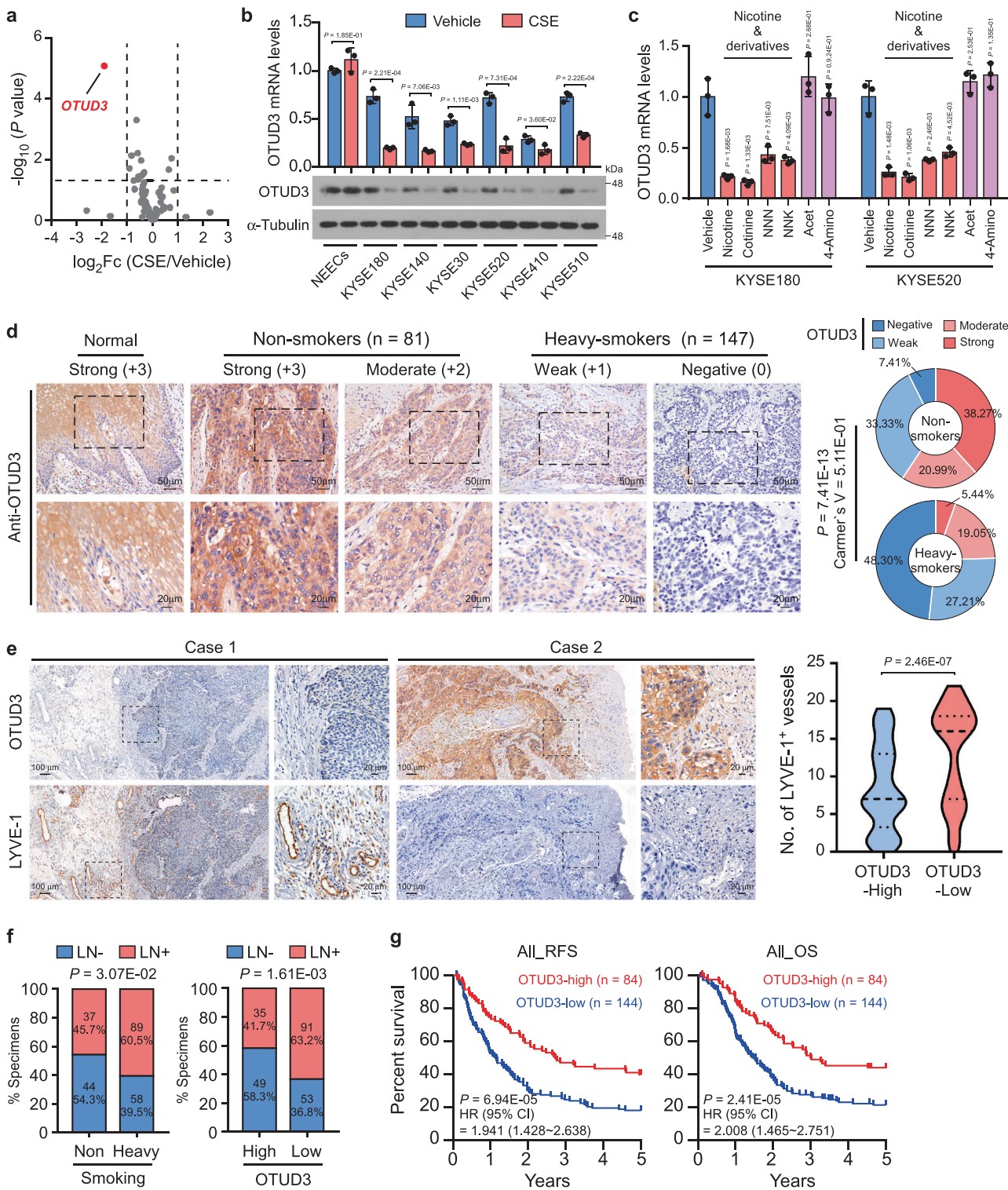

We further assessed the clinical significance of OTUD3 expression in the 228 paraffin-embedded esophageal cancer specimens (Supplementary Table 1). Low OTUD3 expression significantly correlated with clinicopathological characteristics, including tumor size ($P < 0.001$), LN metastasis ($P = 0.002$), pathological grades ($P = 0.04$), and clinical stages ($P < 0.001$; Supplementary Fig. 2a and Supplementary Table 2), indicating a reverse correlation with esophageal cancer malignancy. Notably, OTUD3 was significantly downregulated in the heavy-smoking patients compared to those non-smokers, as well as those in the normal esophageal tissues (Fig. 1d). Significantly, a strong association was found between OTUD3 expression and smoking status[43,44] (Supplementary Fig. 2b), suggesting that OTUD3 expression was downregulated in the current smokers and could be recovered by quitting smoking. These findings indicate that OTUD3 is regulated by smoking in esophageal cancer.

Intriguingly, LYVE-1 staining showed that the number of lymphatic vessels was robustly increased in esophageal tumors with low OTUD3 expression (Fig. 1e). Moreover, correlation analysis indicated that the smoking behavior and low OTUD3

**Fig. 1 OTUD3 downregulation by nicotine correlates with LN metastasis. a** KYSE180 cells were treated with vehicle DMSO or cigarette smoke extract (CSE) (100 μg/ml) for 24 h and then subjected to RNA-seq analysis. A volcanic plot showed the fold changes and P value (DESeq2 Likelihood Ratio Test) of deubiquitinase (DUB) genes under CSE treatment. **b** qRT-PCR and western blot analysis of OTUD3 in the normal esophageal epithelial cell (NEEC) and six esophageal cancer cell lines with vehicle or CSE treatment for 24 h. Immunoblots are representative of three biological replicates. **c** qRT-PCR analysis of OTUD3 mRNA in KYSE180 and KYSE520 cells treated with the indicated tobacco carcinogens. **d** Representative images of OTUD3 IHC staining in normal (*n* = 5) and esophageal cancer specimens from non-smoking (*n* = 81) and heavy-smoking (*n* = 147) patients. The distribution of OTUD3 staining between non-smokers and heavy smokers was compared. Two-sided $\chi^2$ test and Cramer's V were used to evaluate the correlation. Scale bars: 50 μm; insets: 20 μm. **e** The representative images and quantification of LYVE-1-positive lymphatic vessels in OTUD3-low or OTUD3-high specimens. The dotted lines represent the first quartile, median, and third quartile of lymphatic vessel number. Two-sided Student's *t* test was used for statistical analysis. Scale bars: 100 μm; insets: 20 μm. **f** Correlation analyses between LN metastasis status and smoking behavior or OTUD3 expression in patient specimens. Two-sided $\chi^2$ test was used. **g** Relapse-free survival (RFS) and overall survival (OS) analysis in esophageal cancer patients stratified by low and high OTUD3 expression (*n* = 228, log-rank test) were shown. HR hazard ratio. Each error bar in **b**, **c** represents the mean ± SD of three biological replicates. Two-sided Student's *t* test was used for statistical analysis. Source data are provided as a Source data file.

expression were significantly associated with LN metastasis of esophageal cancer (Fig. 1f and Supplementary Table 2). Importantly, low expression of OTUD3 significantly predicted poorer tumor relapse-free survival (RFS) and overall survival (OS) in esophageal cancer patients (Fig. 1g), and this effect was more significant in the heavy-smoking patients than those in non-smokers (Supplementary Fig. 2c, d). Therefore, these results suggest that nicotine-mediated OTUD3 downregulation might contribute to lymphatic metastasis in the smoking esophageal cancer patients, leading to poor clinical outcomes.

As the upstream regulator of OTUD3, we found that nuclear FOXO1 expression positively correlated with OTUD3 expression in esophageal cancer tissues and was significantly associated with the smoking behavior of patients (Supplementary Fig. 2e, f and Supplementary Table 3). Similarly, low nuc-FOXO1 expression performed better in predicting RFS and OS in heavy-smoking patients than in the non-smokers (Supplementary Fig. 2g, h).

**OTUD3 abrogates nicotine-induced lymphatic metastasis**. Interestingly, direct nicotine administration in LECs had no pro-lymphangiogenic effects (Supplementary Fig. 3a, b), suggesting that nicotine might promote tumor-induced lymphangiogenesis. Strikingly, the conditioned medium (CM) derived from nicotine-treated esophageal cancer cells activated the VEGFR downstream signaling transduction in LECs (Fig. 2a and Supplementary Fig. 3c, d). Nicotine-induced CM promoted robust lymphangiogenesis as indicated by the increased migration and tube formation of LECs (Fig. 2b, c). Importantly, restoration of OTUD3 abrogated nicotine-induced VEGFR signaling activation and lymphangiogenesis in vitro (Fig. 2a–c and Supplementary Fig. 3d). By contrast, the CM from OTUD3-silencing esophageal cancer cells significantly increased the migration and tube formation of LECs (Supplementary Fig. 3e–g).

We further investigated the role of nicotine in lymphatic metastasis in vivo (Fig. 2d). Briefly, mice were randomly divided into two groups (*n* = 12/group) and subcutaneously (s.c., daily) administered with vehicle or nicotine. Administration of nicotine in mice was validated by the serum levels of cotinine, the metabolite of nicotine (Supplementary Fig. 3h). Each group was further equally divided and injected with control or OTUD3-overexpressing KYSE180 cells at the footpad. After a month of inoculation, mice were euthanized. The primary footpad tumors were subjected to lymphangiogenesis examination, while metastasis in LNs was analyzed. Remarkably, nicotine promoted intra- and peri-tumoral lymphangiogenesis in vivo as indicated by dual-immunofluorescence (IF) and immunohistochemical (IHC) staining of LEC markers LYVE-1[45] and podoplanin (PDPN)[46] (Fig. 2e, f). Moreover, the administration of nicotine promoted LN metastasis as indicated by the volumes of LNs and the number of metastatic KYSE180 cells in LNs (Fig. 2g, h).

Additionally, the relative mRNA ratio of human HPRT1 to mouse glyceraldehyde 3-phosphate dehydrogenase (GAPDH)[47] indicated a higher proportion of colonized tumor cells in LNs (Fig. 2i). Notably, ectopic expression of OTUD3 robustly prohibited the lymphangiogenesis and LN metastasis induced by nicotine (Fig. 2e–i), suggesting that reduction of OTUD3 was indispensable for nicotine-induced lymphatic metastasis in esophageal cancer.

**Nicotine upregulates VEGF-C by decreasing OTUD3 and ZFP36**. VEGF-C and VEGF-D are the most well-established pro-lymphangiogenic factors[16]. Interestingly, we found that nicotine specifically increased the VEGF-C mRNA levels in a time-dependent manner but did not affect VEGF-D expression (Fig. 3a and Supplementary Fig. 4a). However, VEGF-C mRNA was retained at low levels and not affected by nicotine in OTUD3-overexpressing cells, indicating that repression of OTUD3 was requisite to VEGF-C upregulation (Fig. 3a). Indeed, OTUD3 silencing robustly increased the mRNA and secreted protein levels of VEGF-C (Fig. 3b and Supplementary Fig. 4b). Notably, OTUD3 showed no effects on the *VEGF-C* promoter activity (Supplementary Fig. 4c). These data suggested that the observed VEGF-C mRNA upregulation might not be caused via direct transcription but possibly in a post-transcriptional manner. As expected, the half-life of VEGF-C mRNA was prolonged in OTUD3-silencing cells but shortened in the OTUD3-overexpressing cells (Fig. 3c), indicating that OTUD3 regulated VEGF-C mRNA decay.

The mRNA decay requires the involvement of RBPs[25]. Intriguingly, mass spectrometric (MS) analysis of OTUD3-interacting proteins concentrated on two RBPs, DDX3X and ZFP36 (Fig. 3d and Supplementary Fig. 4d). Of note, ZFP36 silencing, but not DDX3X, significantly increased VEGF-C mRNA expression (Fig. 3e). In contrast, overexpression of ZFP36 reduced VEGF-C mRNA levels in a dose-dependent manner (Fig. 3f). Notably, nicotine potently decreased ZFP36 expression in esophageal cancer cell lines and tumors, and these effects were abrogated by OTUD3 overexpression, suggesting that OTUD3 mediated the nicotine-induced ZFP36 downregulation (Fig. 3g, h). Likewise, ectopic expression of OTUD3 potently impaired nicotine-mediated VEGF-C protein and mRNA upregulation in tumors (Fig. 3h and Supplementary Fig. 4e). These data reveal that ZFP36 may be the select RBP downstream of OTUD3 to regulate VEGF-C mRNA decay.

OTUD3 did not regulate the mRNA expression of ZFP36 (Supplementary Fig. 4f). Considering that OTUD3 can stabilize proteins, we speculated that OTUD3 might inhibit ZFP36 degradation. As expected, OTUD3 increased ZFP36 protein expression in a dose-dependent manner; however, the enzyme-dead OTUD3$^{C76A}$ mutant had no effect (Fig. 3i). Conversely,

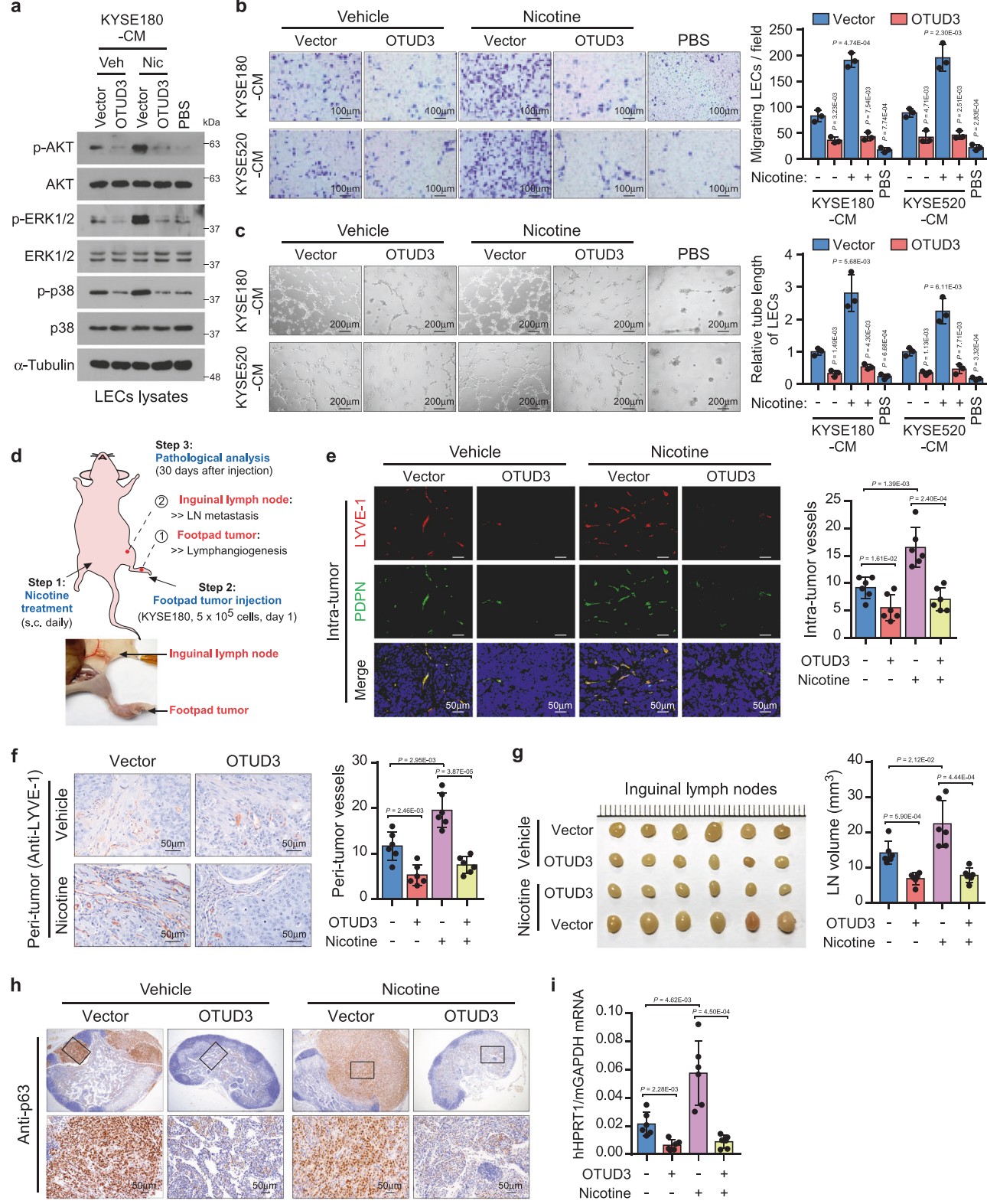

silencing of OTUD3 decreased ZFP36 in esophageal cancer cells, and this effect was abolished under the treatment of proteasome inhibitor MG132 (Fig. 3j). Moreover, OTUD3 potently increased, while silencing of OTUD3 dramatically reduced the half-life of ZFP36 protein (Fig. 3k). These findings indicate that OTUD3 stabilizes the ZFP36 protein.

Notably, the OTUD3$^{C76A}$ mutant showed no significant effects on VEGF-C expression, tumor-induced VEGFR signaling,

migration, and tube formation of human LECs (hLECs; Supplementary Fig. 4g–j), indicating that the catalytic activity of OTUD3 is indispensable for its role in lymphangiogenesis.

**OTUD3 inhibits the K48-linked ubiquitination of ZFP36.** We further investigated whether OTUD3 regulated the ubiquitination of ZFP36. Indeed, endogenous and exogenous reciprocal

**Fig. 2 OTUD3 abrogates nicotine-induced lymphatic metastasis. a** Conditioned medium (CM) was collected from vector or OTUD3-overexpressing esophageal cancer cells cultured with vehicle DMSO or nicotine (2 µM) for 24 h and used to stimulate lymphatic endothelial cells (LECs). LECs were then subjected to western blot analysis of the indicated proteins. PBS was used as a negative control. Immunoblots are representative of three biological replicates. **b** LECs were incubated with the indicated CM and then subjected to transwell migration assays. Representative images of three biological replicates were shown. Scale bars: 100 µm. **c** Tube-formation assays of LECs incubated with the indicated CM. PBS was used as a negative control. Representative images of three biological replicates are shown. Scale bars: 200 µm. **d** Scheme procedure of nicotine administration and lymphatic metastasis examination in the mouse model. Briefly, mice were subcutaneously injected with nicotine (0.75 mg/kg, s.c.). Nicotine administration continued daily and last for an entire month. Vector or OTUD3-overexpressing KYSE180 cells ($5 \times 10^5$) were injected into the footpads of mice 1 day after the first nicotine injection and was defined as day 1. There were four groups ($n = 6$ mice/group). After a month of inoculation, the primary footpad tumors and inguinal lymph nodes (LNs) were subjected to pathological examination. **e** The dual-immunofluorescence (IF) staining of lymphatic endothelial cell markers LYVE-1 and podoplanin (PDPN) was performed in footpad tumor tissue sections. Representative images and quantification of intra-tumoral lymphatic vessels were shown. Scale bars: 50 µm. **f** Representative images and quantification of peri-tumoral lymphatic vessels in footpad tumors as indicated by IHC staining of LYVE-1. Scale bars: 50 µm. **g** Image and volumes of the inguinal lymph nodes (LNs) from each group. **h** Representative images showed metastatic KYSE180 cells in LNs as indicated by the IHC staining of squamous cell carcinoma marker p63. Scale bars: 50 µm. **i** qRT-PCR analysis of human HPRT1 relative to mouse GAPDH in the LNs from each group. The ratio indicated the proportion of metastatic cells. Each error bar in **b**, **c** represents the mean ± SD of three biological replicates. Each error bar in **e–g**, **i** represent the mean ± SD derived from tumor mouse models ($n = 6$ mice/group). Two-sided Student's $t$ test was used for all panels. Source data are provided as a Source data file.

immunoprecipitation (IP) assays revealed that OTUD3 interacted with ZFP36 (Fig. 4a, b). Notably, nicotine decreased the interactions between OTUD3 and ZFP36 as indicated by the IP and proximity ligation assays (PLAs; Fig. 4c and Supplementary Fig. 5a). The PLAs and dual-IF staining revealed that OTUD3 and ZFP36 colocalized in the cytoplasm of esophageal cancer cells (Fig. 4c and Supplementary Fig. 5b). Further truncation IP assays revealed that the OTU domain of OTUD3 and N-terminal of ZFP36 were essential for their reciprocal interactions (Fig. 4d). Moreover, in vitro binding assays using purified proteins showed that OTUD3 interacted with ZFP36 in a dose-dependent manner, suggesting that their interaction was direct (Fig. 4e).

Remarkably, nicotine-mediated OTUD3 downregulation robustly increased the polyubiquitination of ZFP36, and this effect was abolished by OTUD3 re-expression (Fig. 4f). Consistently, overexpression of OTUD3 diminished, while the silencing of OTUD3 promoted the polyubiquitination of ZFP36 (Fig. 4g). Although OTUD3[C76A] still interacted with ZFP36, it lost the ability to deubiquitinate ZFP36 (Fig. 4g and Supplementary Fig. 5c). To exclude the possibility that the ubiquitination signals observed by immunoblotting might be contributed by additional ZFP36-associated proteins, the IP assays were also performed under denaturing conditions. Notably, similar results were observed, indicating that OTUD3 specifically regulated the polyubiquitination of ZFP36 (Supplementary Fig. 5d, e). Moreover, via transfection of Ub-wt, Ub-K48-only, or Ub-K63-only, we found that OTUD3 removed K48- but not K63-linked polyubiquitination of ZFP36 (Fig. 4h and Supplementary Fig. 5f). Interestingly, nicotine was found to inhibit the activity of proteasomes[44], suggesting that nicotine may generally inhibit protein degradation post-ubiquitin modifications, which might explain that nicotine further induced an increase in ZFP36 ubiquitination in presence of transfected OTUD3 (Fig. 4f). Nevertheless, nicotine might also regulate other effectors for ZFP36 ubiquitination, which remains to be further explored.

The E3 ligase FBXW7 was recently identified to induce K48-linked polyubiquitination and degradation of ZFP36[48]. Indeed, FBXW7 promoted the ubiquitination and degradation of ZFP36 in esophageal cancer cells (Fig. 4i, j). Significantly, OTUD3 robustly impaired FBXW7-mediated ZFP36 ubiquitination and degradation in an enzyme-dependent manner (Fig. 4i, j). Moreover, silencing of FBXW7 inhibited polyubiquitination of ZFP36, increased ZFP36 protein expression, and reduced VEGF-C expression in OTUD3-silencing esophageal cancer cells (Supplementary Fig. 5g–i), suggesting that downregulation of FBXW7 rescues the effects of OTUD3 depletion in ZFP36 and

VEGF-C expression. Taken together, these findings reveal that OTUD3 inhibits the ubiquitination and degradation of ZFP36.

The dynamic of polyubiquitination is balanced both by E3 ligase-induced ubiquitination and DUBs-mediated deubiquitination. Interestingly, the FBXW7 E3 ligase was not affected by CSE or nicotine in esophageal cancer cells (Fig. 1a and Supplementary Fig. 5j). These findings suggest that nicotine decreases ZFP36 expression by downregulating OTUD3, but not through alteration of FBXW7, leading to VEGF-C elevation.

**Nicotine inhibits VEGF-C mRNA decay.** The 3'UTRs of VEGF family genes are different (Supplementary Fig. 6a). Notably, except for VEGF-B, VEGF-A, VEGF-C, and VEGF-D all contained AREs (Fig. 5a and Supplementary Fig. 6a), suggesting that their mRNA levels might be regulated via decay. Indeed, VEGF-A was reported as a direct target of ZFP36 in angiogenesis (blood vessels)[33]; however, the mRNA decay regulation of VEGF-C and VEGF-D in lymphangiogenesis (lymphatic vessels) remains unclear. Subsequently, RNA immunoprecipitation (RIP) assays revealed that ZFP36 bound explicitly with the VEGF-C but not with VEGF-D or ACTB mRNA (Fig. 5b and Supplementary Fig. 6b). Moreover, RNA pulldown assays indicated that ZFP36 interacted with the 3'UTR, but neither with 5'UTR nor the coding sequence (CDS) region of VEGF-C mRNA (Fig. 5c). Mutation at ARE or competition using the non-biotin-labeled 3'UTR substantially diminished the interaction between ZFP36 and VEGF-C 3'UTR (Fig. 5d, e). These observations indicate that ZFP36 interacts with the VEGF-C 3'UTR via the conserved ARE. Notably, the specificity of mRNA decay is determined both by cis-acting ARE sequence and trans-acting RBPs, including ZFP36, AUF-1, KSRP, HuR, TIA-1, and TIAR[24,25]. We speculated that the mRNA decay of VEGF-D might be regulated by other RBPs, which remains to be identified in the future.

Once bound with substrate mRNA, ZFP36 recruits the CCR4-NOT deadenylase complex for subsequent mRNA degradation[27]. As expected, Our RIP assays indicated that the enrichment of ZFP36 and CNOT1 (CCR4-NOT deadenylase complex subunit 1) on VEGF-C mRNA was significantly reduced in ZFP36-silencing KYSE180 cells (Fig. 5f and Supplementary Fig. 6c). The amount of retrieved VEGF-C mRNA was related to recovered ZFP36 protein in the RIP assays (Fig. 5b, f), further suggesting that ZFP36 binds with the VEGF-C mRNA. Consequently, ZFP36 overexpression induced rapid degradation of VEGF-C mRNA, while VEGF-C mRNA's half-life was prolonged by ZFP36 silencing (Fig. 5g). Likewise, ZFP36 significantly reduced the luciferase activities conjugated with VEGF-C 3'UTR, but not

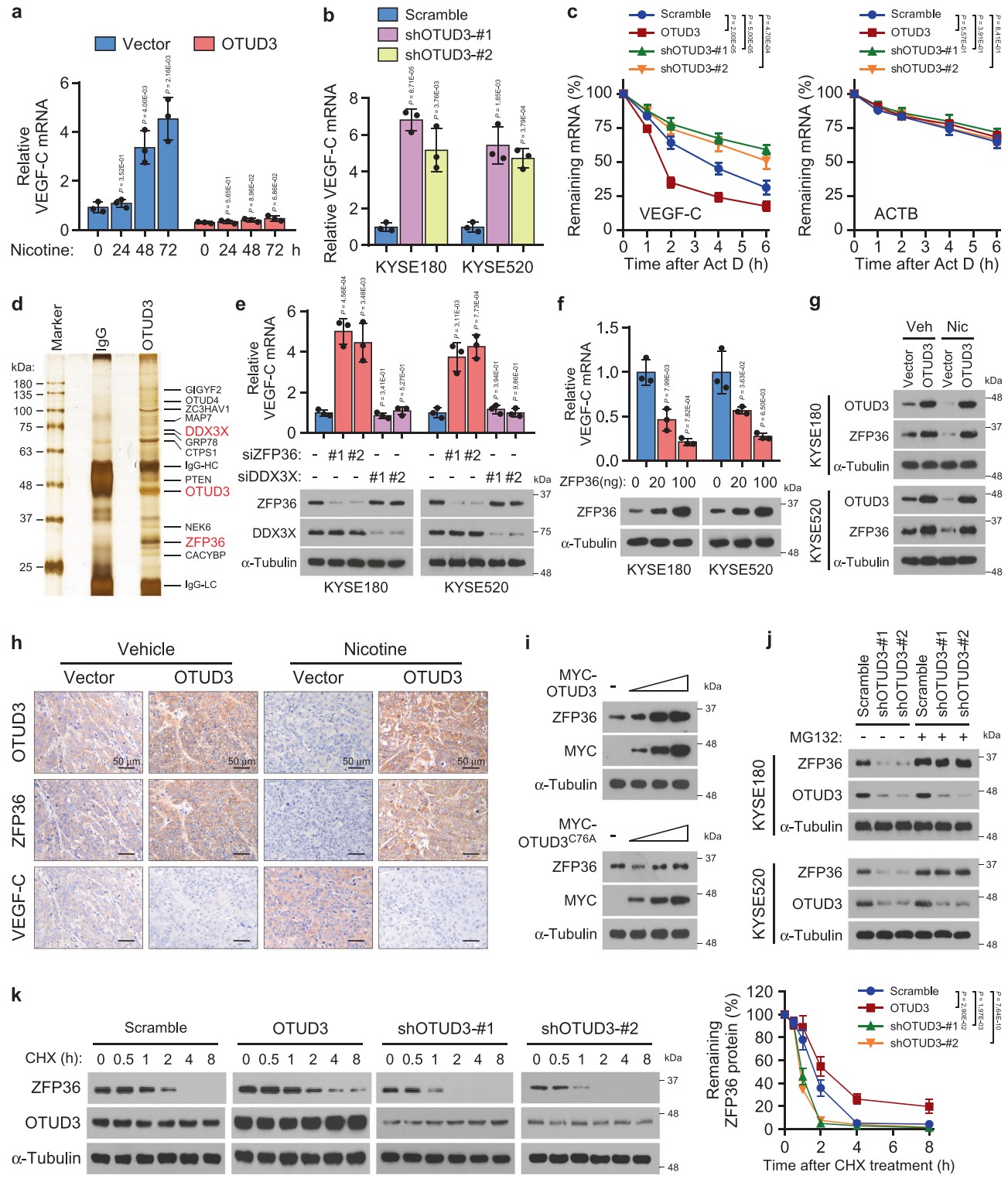

those with ARE-mutant (Fig. 5h). Moreover, ZFP36 regulated the half-life of luciferase mRNA conjugated with VEGF-C 3'UTR, but not the one with ARE-mutant (Supplementary Fig. 6d). These findings reveal that ZFP36 induces VEGF-C mRNA decay.

Notably, nicotine remarkably reduced the enrichment of ZFP36 on VEGF-C 3'UTR and increased the activity of luciferase-VEGF-C-3'UTR (Fig. 5i–l). Thus, nicotine promoted the stability of VEGF-C mRNA (Fig. 5m). Significantly, over-expression of OTUD3 or ZFP36 abrogated nicotine's effects on VEGF-C mRNA stabilization (Fig. 5i–m). These findings suggest

that nicotine inhibits VEGF-C mRNA decay via the down-regulation of OTUD3 and ZFP36.

**Induction of VEGF-C mRNA decay abrogates lymphatic metastasis**. We further evaluated the potential of ZFP36-induced VEGF-C mRNA decay in lymphatic metastasis suppression. We overexpressed ZFP36 in nicotine-treated or OTUD3-silencing esophageal cancer cells (Supplementary Fig. 7a). As expected, ZFP36 substantially reduced the VEGF-C mRNA expression in all these cell models, indicating that decay inhibition was essential

**Fig. 3 Nicotine upregulates VEGF-C by decreasing OTUD3 and ZFP36. a** Vector or OTUD3-overexpressing KYSE180 cells were treated with vehicle DMSO or nicotine for 24, 48, and 72 h, respectively. RNA was extracted and then subjected to qRT-PCR analysis of VEGF-C. **b** qRT-PCR analysis of VEGF-C in the scramble and OTUD3 silencing esophageal cancer cells. **c** Scramble, OTUD3-overexpressing, or OTUD3-silencing KYSE180 cells were treated with Act D (5 μg/ml). RNA was isolated at the indicated time points and then subjected to qRT-PCR analysis of VEGF-C and ACTB. The half-lives of mRNAs were traced by calculating the remaining mRNA levels relative to the untreated cells. **d** Immunoprecipitation of OTUD3-interacting proteins was subjected to SDS-PAGE, silver-stained, and identified by mass spectrometry (MS). **e** qRT-PCR analysis of VEGF-C in esophageal cancer cells transfected with NC, ZFP36 siRNAs, or DDX3X siRNAs. The efficiency of ZFP36 and DDX3X knockdown was indicated. **f** qRT-PCR analysis of VEGF-C in esophageal cancer cells transfected with increasing dose of ZFP36-expressing plasmid. ZFP36 expression was validated by western blot analysis. **g** Western blot analysis of OTUD3 and ZFP36 in the indicated cells. Immunoblots are representative of three biological replicates. **h** Representative IHC staining images of OTUD3, ZFP36, and VEGF-C in the indicated footpad tumors. Scale bars: 50 μm. **i** Western blot analysis of ZFP36 and MYC-OTUD3 in KYSE180 cells transfected with increasing MYC-OTUD3 or MYC-OTUD3$^{C76A}$. Immunoblots are representative of three biological replicates. **j** Western blot analysis of ZFP36 and OTUD3 in the scramble and OTUD3-silencing esophageal cancer cells with or without MG132 treatment (10 μM). Immunoblots are representative of three biological replicates. **k** Scramble, OTUD3-overexpressing, and OTUD3-silencing KYSE180 cells were treated with 100 μg/ml cycloheximide (CHX). Proteins were collected at the indicated time points and then immunoblotted with anti-ZFP36 and anti-OTUD3 antibodies. Quantification of ZFP36 protein levels was determined by normalizing to α-tubulin protein. Each error bar in **a–c**, **e**, **f**, **k** represents the mean ± SD of three biological replicates. Two-sided Student's *t* test (**a**, **b**, **e**, **f**), or One-way repeated-measures ANOVA test (**c**, **k**) was used for statistical analysis. Source data are provided as a Source data file.

for the robust VEGF-C production (Fig. 6a). Consistently, ZFP36 overexpression potently inhibited tumor-induced lymphangiogenesis induced by nicotine or OTUD3 depletion, as indicated by LECs' migration and tube formation (Fig. 6b, c). Strikingly, ZFP36 strongly reduced VEGF-C expression and the number of lymphatic vessels in the footpad tumors (Fig. 6d–g and Supplementary Fig. 7b, c). Consequently, ZFP36 abrogated the LN metastasis induced by nicotine or OTUD3 silencing (Fig. 6h–j). All tumor cells highly expressed the squamous cell carcinoma marker p63 in primary KYSE180 tumors (Supplementary Fig. 7d, e). The p63-positive cells in LNs indicated the metastatic KYSE180 cells showing that nicotine and OTUD3 downregulation robustly promoted LN metastasis (Fig. 6i). The relative mRNA ratio of human HPRT1 to mouse GAPDH[47] indicated that ZFP36 strikingly decreased the proportion of colonized tumor cells in LNs (Fig. 6j). Notably, ZFP36 induced inhibitory effects on LN metastasis to an extent below control, suggesting that, besides VEGF-C, ZFP36 might repress LN metastasis via other identified ZFP36 effects such as pro-apoptosis[49]. These findings indicate that ZFP36 inhibits nicotine-induced lymphatic metastasis both in vitro and in vivo.

Notably, we previously found that TBL1XR1 transcriptionally upregulated VEGF-C to promote lymphangiogenesis[20]. We further tested whether ZFP36 overexpression blocked the role of TBL1XR1 (Supplementary Fig. 7f). Significantly, ZFP36 strongly reduced VEGF-C mRNA levels in TBL1XR1-overexpressing esophageal cancer cells, resulting in robust inhibition of TBL1XR1-mediated LEC migration and tube formation (Supplementary Fig. 7g–i). Taken together, these pieces of evidence suggest that induction of VEGF-C mRNA decay might be a promising therapeutic strategy against lymphatic metastasis in esophageal cancer.

**Clinical relevance and study model.** Finally, we assessed the OTUD3/ZFP36/VEGF-C axis's clinical relevance in esophageal cancer specimens. OTUD3 and ZFP36 were potently downregulated, while VEGF-C was elevated in the freshly collected esophageal cancer samples from the heavy-smoking patients compared to those from non-smokers, suggesting a significant relevance of the OTUD3/ZFP36/VEGF-C axis in primary tumors (Fig. 7a).

Furthermore, the expression of ZFP36 and VEGF-C was assessed in patient specimens. Similar to OTUD3, ZFP36 and VEGF-C expression correlated with smoking behavior, clinical stages, and performed better in predicting RFS and OS in heavy-smoking patients than in the non-smokers (Supplementary

Fig. 8a–d and Supplementary Tables 4 and 5). Significantly, the IHC staining and correlation analysis showed that OTUD3 positively correlated with ZFP36 expression levels but was reversely associated with VEGF-C expression levels in the 228 esophageal cancer specimens (Fig. 7b, c and Supplementary Table 2). Notably, the proportions of strong VEGF-C expression were robustly increased in OTUD3-low specimens, while the proportions of negative VEGF-C expression were decreased in the OTUD3-low group. The Cramer's V correlation coefficient of the test was 0.448, indicating a very strong (defined as >0.25)[43] relationship between OTUD3 and VEGF-C expression (Fig. 7c). Importantly, patients with combined low OTUD3 and high VEGF-C expression suffered the worse RFS and OS, especially in the heavy-smoking patients (Fig. 7d–f), further suggesting that the OTUD3/VEGF-C axis was indeed associated with poor clinical outcomes of esophageal cancer.

Taken together, we here show that nicotine-mediated OTUD3 downregulation facilitates ZFP36 protein degradation and inhibits VEGF-C mRNA decay, leading to robust tumor-induced lymphangiogenesis and lymphatic metastasis in esophageal cancer (Fig. 8). This study uncovers a mechanism for nicotine in lymphatic metastasis, indicating the great importance of smoking cessation and suggesting that induction of VEGF-C mRNA decay might be a promising strategy.

## Discussion

Over the past decades, studies indicate that nicotine contributes to the development and progression of lung, esophageal, pancreatic, and other human cancers[50–53]. Various lines of evidence from cell and animal models demonstrated that nicotine can trigger genotoxic effects by increasing reactive oxygen species to drive mutagenesis and carcinogenesis[54–56]. In particular, nicotine is pro-metastatic in the multi-steps of cancer metastasis. Generally, nicotine can bind to nAChRs to activate several oncogenic signaling pathways to facilitate cancer cell invasion, survival, and outgrowth in metastasis[41,57]. Moreover, nicotine reduces the expression of epithelial marker E-cadherin and increases the mesenchymal markers vimentin and fibronectin to elicit epithelial–mesenchymal transition and cell migration[58]. In addition, there is accumulating evidence that nicotine creates tumor-supporting environments by inducing extracellular matrix degradation and/or reorganization and substantial angiogenesis to facilitate tumor survival and spreading[59,60]. The lymphatic vessels serve as the primary route for the dissemination of many cancers[16]; however, the role of nicotine in lymphangiogenesis remains largely unknown. Herein, we found that nicotine had no

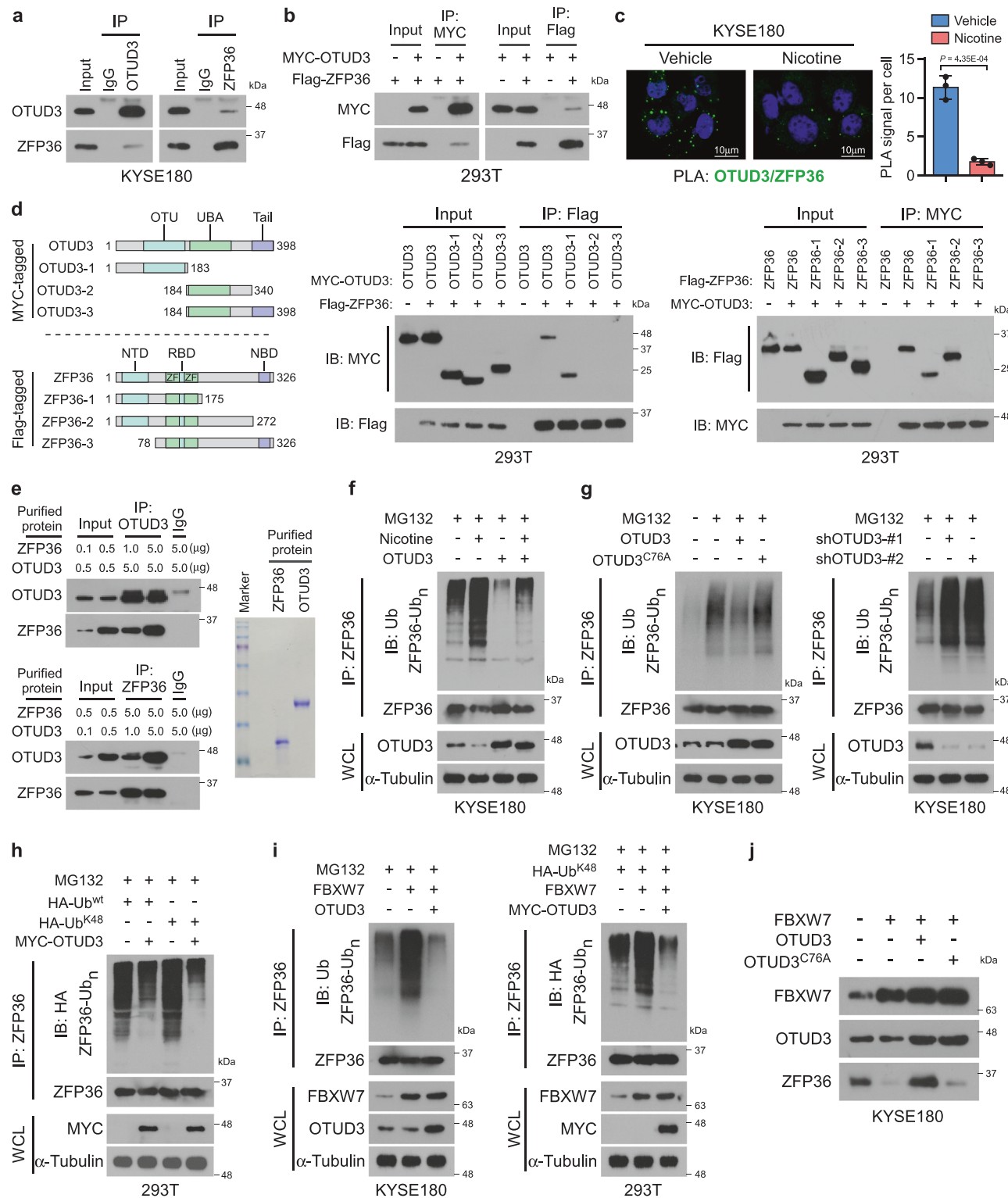

direct impacts on migration and tube formation of hLECs but promoted tumor-induced lymphangiogenesis. Specifically, nicotine activated the PI3K/AKT signaling through the alpha 7 nAChR to inhibit FOXO1 activity and downregulate OTUD3 expression, leading to inhibition of VEGF-C mRNA decay and increase of VEGF-C paracrine secretion in esophageal cancer cells. These findings uncover a mechanism for nicotine in tumor-induced lymphangiogenesis, further suggesting that nicotine can promote tumor metastasis by remodeling the microenvironment.

For esophageal cancer, more than half of new cases and deaths occurred in China[1]. The overall 5-year patient survival rate ranges from 15 to 25%, mainly because patients are at an advanced stage at diagnosis, and the propensity for LN metastasis even when tumors are small[1,61]. The disseminated tumor cells in LNs are the direct cause of tumor relapse post-therapy[5,6]. Although a well-established causative relationship exists between smoking and several epithelial cancers[62], the relation between smoking and the metastatic progression in esophageal cancer is not obscure. In this study, we found

**Fig. 4 OTUD3 inhibits the K48-linked ubiquitination of ZFP36. a** Reciprocal immunoprecipitation (IP) assays of OTUD3 and ZFP36 in KYSE180 cells. **b** The 293T cells were transfected with MYC-OTUD3 and Flag-ZFP36 constructs, followed by IP assays using MYC- or Flag-conjugated agarose. **c** Proximity ligation assay (PLA) indicated the interaction between OTUD3 and ZFP36 in KYSE180 cells treated with vehicle DMSO or nicotine. Representative images of three biological replicates are shown. Scale bars: 10 μm. **d** Schematic illustrations of OTUD3 and ZFP36 truncated constructs are shown. IP assays were performed in 293T cells with the indicated transfections to examine the interactions between OTUD3 and truncated ZFP36 or between ZFP36 and truncated OTUD3. **e** In vitro protein-binding assays with increasing dose of purified OTUD3 and ZFP36 proteins. The purities of OTUD3 and ZFP36 were examined by SDS-PAGE and Coomassie Blue staining. **f** Vector or OTUD3-overexpressing KYSE180 cells were treated with or without nicotine. Cells were treated with MG132 (10 μM) for 6 h before harvest. Lysates were then immunoprecipitated with anti-ZFP36 antibody, followed by immunoblotting with anti-Ub and anti-ZFP36 antibodies to examine the ubiquitination levels. WCL whole-cell lysate. **g** KYSE180 with OTUD3, OTUD3$^{C76A}$ overexpression, or OTUD3 depletion were treated with MG132 (10 μM) and then subjected to ZFP36 ubiquitination analysis. **h** The 293T cells were transfected with HA-Ub$^{wt}$, HA-Ub$^{K48}$, and MYC-OTUD3. Lysates were then subjected to ZFP36 ubiquitination analysis. **i** Analysis of endogenous or K48-linked ubiquitination of ZFP36 in KYSE180 cells transfected with FBXW7 or FBXW7 plus OTUD3. **j** Western blot analysis of FBXW7, OTUD3, and ZFP36 in the indicated KYSE180 cells. OTUD3, but not OTUD3$^{C76A}$, abrogated FBXW7-mediated ZFP36 downregulation. Immunoblots in this figure are representative of three biological replicates. Each error bar in **c** represents the mean ± SD of three biological replicates. Two-sided Student's *t* test was used. Source data are provided as a Source data file.

that smoking significantly correlated with LN metastasis status in clinical specimens. Moreover, using the mouse LN metastasis model, we demonstrated that nicotine exposure potently promoted esophageal cancer cells' lymphatic metastasis. These findings establish a promotive role of nicotine addiction in LN metastasis of esophageal cancer. Notably, e-cigarettes are becoming popular in many areas as an alternative to cigarette smoking. Although there is no conclusive evidence that vaping causes esophageal cancer, most e-cigarettes contain nicotine in varying concentrations and certain studies suggest that vaping could potentially increase nicotine-related carcinogens in vivo[63]. Therefore, it would be of great significance to investigate whether OTUD3 is downregulated and plays a critical role in e-cigarette-using esophageal cancer patients in the future.

Notably, the metastatic tropism of LN spread is believed to be an active rather than a passive process. Apart from migration and invasion, tumors can induce intra- and peri-tumoral lymphangiogenesis, providing numerous lymphatic vessels to facilitate metastasis[16]. Significantly, our previous and current studies showed that esophageal tumors that had higher indexes of intra- and peri-tumoral lymphatic vessels were much more prone to exhibit LN metastasis[20]. These data further strengthen the critical role of tumor-induced lymphangiogenesis in esophageal cancer progression. Notably, both studies identified that VEGF-C, rather than VEGF-D, was the more crucial pro-lymphangiogenic factor, suggesting that VEGF-C might be a promising therapeutic target in esophageal cancer.

Although the transcriptional regulation of VEGF-C is well studied[64], little is known about its post-transcriptional regulation. Notably, accumulating evidence reveals that balancing gene transcription and its mRNA decay is critical for cytokine productions[23]. The decay of mRNA is controlled by the *cis*-acting ARE sequence and *trans*-acting RBPs[24,25]. ZFP36 was identified as a suppressor of multiple inflammatory cytokines and was downregulated in many human cancers[23]. Notably, a reverse relation of ZFP36 and VEGF-C expression was found in uremic rats with peritoneal dialysis; however, the underlying regulation remains unclear. In this study, we found that the VEGF-C mRNA had a conserved ARE in the 3′UTR. ZFP36 interacted with VEGF-C mRNA via the ARE and induced its rapid decay by recruiting the RNA-degrading complex. Thus, loss of ZFP36 expression mediated nicotine-induced VEGF-C mRNA upregulation. Notably, ZFP36 abrogated the transcriptional or post-transcriptional upregulation of VEGF-C to retain it at a low level, resulting in substantial repression of lymphatic metastasis of esophageal cancer cells. Thus, these findings uncover a post-transcriptional mechanism for VEGF-C upregulation in cancer and suggest that induction of mRNA decay by ZFP36 might be a promising strategy for VEGF-C blockage.

OTUD3 was reported to play a context-dependent role in cancers. It suppresses tumorigenesis in breast, colon, liver, brain, and cervical cancer by stabilizing PTEN and p53, while promoting lung tumorigenesis by stabilizing GRP78[36–38,65]. For esophageal cancer, cigarette smoking is recognized as a significant prognostic factor, contributing to tumor malignant progression[2]. In this study, we found that OTUD3 was highly expressed in NEECs and tissues. Notably, OTUD3 was substantially decreased under nicotine exposure in esophageal cancer cells and heavy-smoking patient specimens. OTUD3 stabilized the tumor-suppressive ZFP36 protein and induced rapid mRNA decay of VEGF-C. Thus, OTUD3 downregulation was essential for nicotine-induced lymphatic metastasis, leading to earlier tumor relapse and shorter patient survival, especially in heavy smokers. These observations further suggest that OTUD3 plays a tumor-suppressive role in esophageal cancer. Notably, OTUD3 plays pleiotropic functions by inducing different patterns of deubiquitination on various substrate proteins[39,40]. In this study, apart from ZFP36, we identified many other potential OTUD3-interacting proteins. It would be of great significance to further explore whether these proteins contribute to the tumor-suppressive role of OTUD3 in esophageal cancer.

## Methods

**Cells**. The human esophageal cancer cell lines, including KYSE180, KYSE520, KYSE140, KYSE30, KYSE410, and KYSE510, were obtained from Deutsche Sammlung von Mikroorganismen und Zellkulturen (DSMZ, Braunschweig, Germany), the German Resource Centre for Biological Material[66]. 293T cells were obtained from the Cell Bank of Shanghai Institutes of Biological Sciences (Shanghai, China). Cell lines were authenticated by short tandem repeat fingerprinting. Cells were grown in Dulbecco's Modified Eagle Medium supplemented with 10% fetal bovine serum, penicillin/streptomycin, hydrocortisone, insulin, HEPES, and L-glutamine. The human NEECs were purchased from ScienCell (Carlsbad, CA, USA) and propagated in the recommended serum-supplemented medium. All cells were maintained in 5% $CO_2$ at 37 °C.

**Patient information and tissue specimens**. This study used 228 paraffin-embedded esophageal cancer patient specimens that had been clinically and histopathologically diagnosed at the Sun Yat-sen University Cancer Center from 2008 to 2015. Adjuvant therapy was normally conducted in esophageal cancer patients after tumor resections, with only some early-stage tumors not receiving it. The specimens were derived from 81 non-smoking patients who have never smoked before diagnosis and 147 heavy-smoking patients defined by an overtime smoking history estimated by pack-years no less than 20 (PY > 20)[44]. The heavy-smoking patients were divided into the former and current categories as previously reported[44], in which former smokers were defined as those who quit smoking at least 1 year before surgery. The clinicopathological characteristics are summarized in Supplementary Table 1. Freshly collected esophageal cancer tissues derived from six non-smokers and six heavy smokers were frozen and stored in liquid nitrogen until use. Ethics approval (#GZR2016-111) was obtained from the Institutional Research Ethics Committee of Sun Yat-sen University Cancer Center to use the clinical specimens for research purposes. Written informed consent was obtained from patients, and the study was carried out following the Declaration of Helsinki.

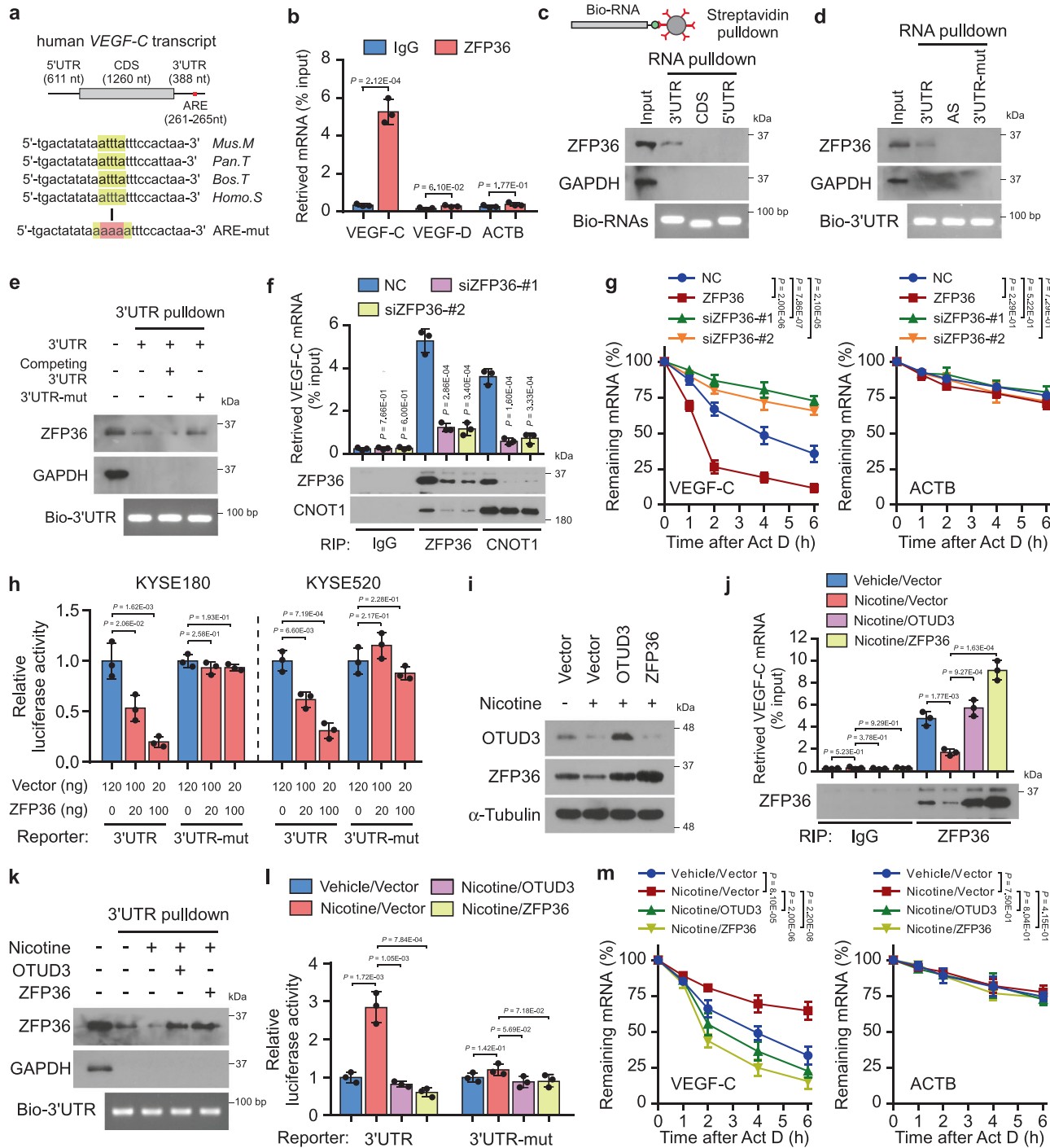

**Immunohistochemistry**. IHC staining was performed on the 228 paraffin-embedded esophageal cancer tissue sections using anti-OTUD3 (Sigma-Aldrich, HPA028544), anti-LYVE-1 (Sigma-Aldrich, HPA042953), anti-ZFP36 (Merck Millipore, ABE285), anti-VEGF-C (R&D, AF752), anti-FOXO1 (Cell Signaling Technology, #2880), and anti-p63 (Sigma-Aldrich, SAB5600140) antibodies. In brief, paraffin-embedded specimens were cut into 4-μm sections and baked at 65 °C for 30 min. The sections were deparaffinized with xylenes and rehydrated. Sections were then submerged into EDTA antigenic retrieval buffer and microwaved for antigenic retrieval. Samples were treated with 3% hydrogen peroxide in methanol to quench the endogenous peroxidase activity, followed by incubation with 1% bovine serum albumin to block nonspecific binding, and then incubated with primary antibodies overnight at 4 °C. After washing, the tissue sections were treated with a biotinylated secondary antibody, followed by further incubation with streptavidin–horseradish peroxidase complex (Zsbio, BJ, China). Finally, the sections were immersed in 3-amino-9-ethyl carbazole and counterstained with 10% Mayer's hematoxylin, dehydrated, and mounted in Crystal Mount.

The staining results were evaluated and scored by two independent pathologists, who were blinded to the clinical outcome. The staining of OTUD3, ZFP36, and

VEGF-C was graded with four scores, strong +3, moderate +2, weak +1, and negative 0. Specimens with scores +3 and +2 were defined as high expression, while the others scored as +1 or 0 were defined as low expression. On the other hand, specimens with >10% nuclear FOXO1 expression were defined as nuc-FOXO1-pos, and specimens with ≤10% nuclear FOXO1 expression were nuc-FOXO1-neg. The number of lymphatic vessels was measured by counting LYVE-1-positive vessels as previously reported[67]. Under a magnification of ×200 microscope, 5 view fields were randomly selected for each slice, and the average number of LYVE-1-positive lymphatic vessels was counted and the whole number was kept. The primary footpad tumors or LNs were stained with the squamous cell carcinoma marker p63 to determine the KYSE180 cells.

**Chemicals**. CSE was prepared as previously described[68,69]. Briefly, cigarette smoke was collected by a vacuum machine into a container frozen with liquid nitrogen, then dissolved in dimethylsulfoxide (DMSO) and stored at −80 °C. Smoke condense was used at a concentration of 100 μg/ml relevant to the human exposure

**Fig. 5 Nicotine inhibits VEGF-C mRNA decay. a** The VEGF-C 3'UTR contained a conserved AU-rich element (ARE) among species. **b** RNA immunoprecipitation (RIP) assays were performed in KYSE180 cells using an anti-ZFP36 antibody. Retrieved RNA was then subjected to qRT-PCR analysis of VEGF-C, VEGF-D, and ACTB. IgG was used as a negative control. **c** In vitro-transcribed and biotin-labeled VEGF-C 5'UTR, CDS, and 3'UTR RNAs were incubated with KYSE180 lysates, followed by streptavidin pulldown and detection of ZFP36. GAPDH was used as a negative control. Immunoblots are representative of three biological replicates. **d** Western blot analysis of ZFP36 following the streptavidin pulldown of biotin-labeled VEGF-C 3'UTR, 3'UTR-ARE-mut, or 3'UTR-AS in KYSE180 lysates. Immunoblots are representative of three biological replicates. **e** Streptavidin pulldown of biotin-labeled VEGF-C 3'UTR with or without competing 3'UTR (unlabeled) or bio-3'UTR-mut. The interaction of ZFP36 protein was examined by western blot. Immunoblots are representative of three biological replicates. **f** RIP assays of ZFP36 and CNOT1 followed by the qRT-PCR analysis of VEGF-C in KYSE180 cells transfected with NC or ZFP36 siRNAs. Enrichment of immunoprecipitated ZFP36 and CNOT1 proteins was indicated. **g** The half-life of VEGF-C mRNA was traced in the indicated KYSE180 cells. ACTB was used as a negative control. **h** Luciferase reporters of VEGF-C 3'UTR and 3'UTR-mut in KYSE180 cells transfected with increasing ZFP36. **i** Western blot analysis of OTUD3 and ZFP36 in KYSE180 cells with or without nicotine treatment. **j** RIP assays of ZFP36 followed by the qRT-PCR analysis of VEGF-C in the indicated KYSE180 cells. Enrichment of immunoprecipitated ZFP36 proteins was indicated. **k** Streptavidin pulldown of biotin-labeled VEGF-C 3'UTR in the indicated KYSE180 lysates, followed by ZFP36 detection. Immunoblots are representative of three biological replicates. **l** Luciferase reporters of VEGF-C 3'UTR or 3'UTR-mut in the indicated KYSE180 cells. **m** VEGF-C mRNA's half-life was traced in Vector, OTUD3-overexpressing, and ZFP36-overexpressing KYSE180 cells with or without nicotine treatment. Each error bar in **b**, **f–h**, **j**, **l**, **m** represents the mean ± SD of three biological replicates. Two-sided Student's *t* test (**b**, **f**, **h**, **j**, **l**) or one-way repeated-measures ANOVA test (**g**, **m**) was used for statistical analysis. Source data are provided as a Source data file.

situation. Cigarette smoke chemical components, including nicotine, cotinine, $N'$-nitrosonornicotine, and 4-(methylnitrosamino)-1-(3-pyridyl)-1-butanone, were purchased from the National Center for Standard Substances (Beijing) and used at a concentration of 2 μM. Pathway-specific inhibitors, including PI3K inhibitor LY294002, RAF inhibitor LY3009120, and JAK2 inhibitor Fedratinib were obtained from Selleck Chemicals and used at concentrations of 500, 20, and 3 nM, respectively. The proteasome inhibitor MG132 was obtained from Sigma-Aldrich and used at 10 μM.

**Western blot analysis**. Western blot analyses were performed according to a standard protocol using primary antibodies, including anti-p-Akt, anti-Akt, anti-p-ERK1/2, anti-ERK1/2, anti-p-p38, anti-p38 antibodies (Cell Signaling Technology, Danvers, MA, USA), anti-OTUD3 (Sigma-Aldrich, MABS1819M), anti-ZFP36 (Merck Millipore, ABE285), and anti-FBXW7 (Abcam, ab109617). GAPDH (Cell Signaling Technology, #5174) and α-Tubulin (Sigma-Aldrich, T9026) were used as loading controls. Western blot grayscale analyses were performed using the image J 1.42q software. Information on antibodies is provided in Supplementary Table 6. Unprocessed scans of immunoblots are provided as Supplementary Fig. 9.

**Luciferase reporter assays**. In this study, we performed *OTUD3* promoter-luciferase reporter assays to examine the role of FOXO1 on *OTUD3* transcription; on the other hand, we conducted VEGF-C-3'UTR luciferase reporter assays to assess whether ZFP36 induced VEGF-C mRNA decay via its ARE in the 3'UTR. Briefly, 20,000 cells were seeded in triplicate in 48-well plates and allowed to settle for 24 h. One hundred nanograms of reporter plasmid, plus 1 ng of pRL-TK Renilla plasmid (Promega), was transfected into cells using the Lipofectamine 3000 reagent according to the manufacturer's recommendation. Luciferase and Renilla signals were measured 24 h after transfection using the Dual-Luciferase Reporter Assay Kit (Promega) according to a protocol provided by the manufacturer. Relative luciferase activity was calculated as the ratio of luciferase to Renilla signal.

**Tube formation of hLECs**. The hLEC tube formation assay was performed by first pipetting 200 μl Matrigel (BD Biosciences, Bedford, MA) into a 24-well plate, which was then polymerized for 30 min at 37 °C. LECs ($2 \times 10^4$) in 200 μl of CM were added to each well and incubated at 37 °C, 5% $CO_2$ for 12 h. Images were taken using a bright-field microscope (Eclipse 80i, Nikon, Tokyo, Japan) at ×100 magnification. The capillary tubes were quantified by measuring the total length of completed tubule structures.

**Inguinal LN metastasis model**. The inguinal LN metastasis model was performed as previously reported[20]. BALB/c-nude mice (Male, 4–5 weeks old) were purchased and housed in barrier facilities on a 12 h light/dark cycle at temperature 18–22 °C and humidity 50–60%. All experimental procedures were approved by the Institutional Animal Care and Use Committee of Sun Yat-sen University and performed following the Declaration of Helsinki. Briefly, mice were randomly divided into two groups ($n = 12$/group). For nicotine administration in vivo, nicotine (0.75 mg/kg, s.c., daily)[70] was s.c. injected with a 4 mm needle in mice daily and lasted for an entire month. The success of nicotine injection in vivo was validated by the plasma cotinine (the nicotine metabolite) levels using enzyme-linked immunosorbent assay at days 0, 10, 20, and 30. Footpad tumor injection was done 1 day after nicotine injection and was defined as day 1. Each group was further equally divided and injected with control or OTUD3-overexpressing KYSE180 cells ($5 \times 10^5$) at the footpad. After a month of inoculation, mice were euthanized. The primary footpad tumors were paraffin-embedded and subjected to

hematoxylin–eosin and IHC staining. Inguinal LNs were first measured to calculate volumes. LNs were used for quantitative reverse transcription polymerase chain reaction (qRT-PCR) analysis of hHPRT1 and mGAPDH and IHC staining of squamous cell carcinoma marker p63 to determine the proportion of LN-spread KYSE180 cells.

**Protein IP assays**. Cell lysates were prepared from the indicated cells using lysis buffer (150 mM NaCl, 10 mM HEPES, pH 7.4, 1% NP-40). Lysates were then incubated with anti-OTUD3 (Sigma-Aldrich, MABS1819M) or anti-ZFP36 (Merck Millipore, ABE285) antibody and protein G-conjugated agarose (Millipore) overnight at 4 °C. Beads containing affinity-bound proteins were washed 6 times by wash buffer (150 mM NaCl, 10 mM HEPES, pH 7.4, 0.1% NP-40), followed by eluting using 1 M glycine (pH 3.0). Elutes were subjected to MS or western blot analysis. The MS data have been deposited in the iProX database (#PXD028751). Information on peptides and counts for OTUD3-binding proteins analyzed by IP/MS assays is provided as Supplementary Data 1.

**Proximity ligation assays**. Proximity ligation was performed using the Rabbit PLUS and Mouse MINUS Duolink In Situ PLA Kits (Sigma-Aldrich) according to the manufacturer's protocol. Briefly, KYSE180 cells ($5 \times 10^4$) were plated on coverslips and treated with nicotine or vehicle for 24 h. The cells were then washed twice with phosphate-buffered saline (PBS) and fixed with 3.7% formaldehyde in PBS for 15 min at room temperature (RT). Subsequently, cells were washed with TBS (25 mM Tris, 100 mM NaCl, pH 7.4), incubated for 10 min in 50 mM NH4Cl, TBS, washed with TBS, permeabilized for 15 min in 0.1% Triton X-100 in TBS, washed with TBST (0.05% Tween 20 in TBS), and then blocked for 2 h with 0.5% milk powder in TBST in a humidified chamber at RT and incubated overnight at 4 °C with anti-OTUD3 (Sigma-Aldrich, HPA028544) and anti-ZFP36 (Abcam, ab124024) antibodies. After washing with TBST, proximity ligation was performed using the PLA kits (Sigma-Aldrich). Cells were further counterstained with 4,6-diamidino-2-phenylindole (Sigma-Aldrich) to visualize the nuclei. The images were obtained by using laser scanning confocal microscopy (LSM880, Carl Zeiss MicroImaging, Oberkochen, Germany).

**Polyubiquitination analysis**. To analyze the polyubiquitination of ZFP36, we treated cells with 10 μM of the proteasome inhibitor MG132 for 6 h. The cells were washed with PBS, pelleted, and lysed in HEPES buffer (20 mM HEPES, pH 7.2, 50 mM NaCl, 1 mM NaF, 0.5% Triton X-100) plus 0.1% sodium dodecyl sulfate (SDS), 10 μM MG132, and protease-inhibitor cocktail. The lysates were centrifuged to obtain cytosolic proteins and incubated with anti-ZFP36 (Merck Millipore, ABE285) overnight. For denaturing conditions, cells were first denatured in lysis buffer containing 1% SDS in conjunction with heat inactivation at 100 °C for 10 min. The resulting sample is diluted tenfold in HEPES buffer to reduce the SDS concentration before the addition of an antibody for IP. The lysates were then pulled down with agarose beads. The beads were washed six times with HEPES buffer and then eluted with 200 μl of 1 M glycine (pH 3.0). The proteins were boiled in SDS-polyacrylamide gel electrophoresis sample buffer for 5 min and analyzed by immunoblotting with an anti-Ub antibody (Cell Signaling Technology, #3936).

**RNA IP (RIP) assays**. This study performed RIP assays to examine the interactions between ZFP36 and VEGF-C mRNA in esophageal cancer cells. Briefly, cells lysed in the lysis buffer (20 mM Tris-Cl, pH 8.0, 10 mM NaCl, 1 mM EDTA, 0.5% NP-40) supplemented with RNasin (Promega). Lysates were then pulled down with

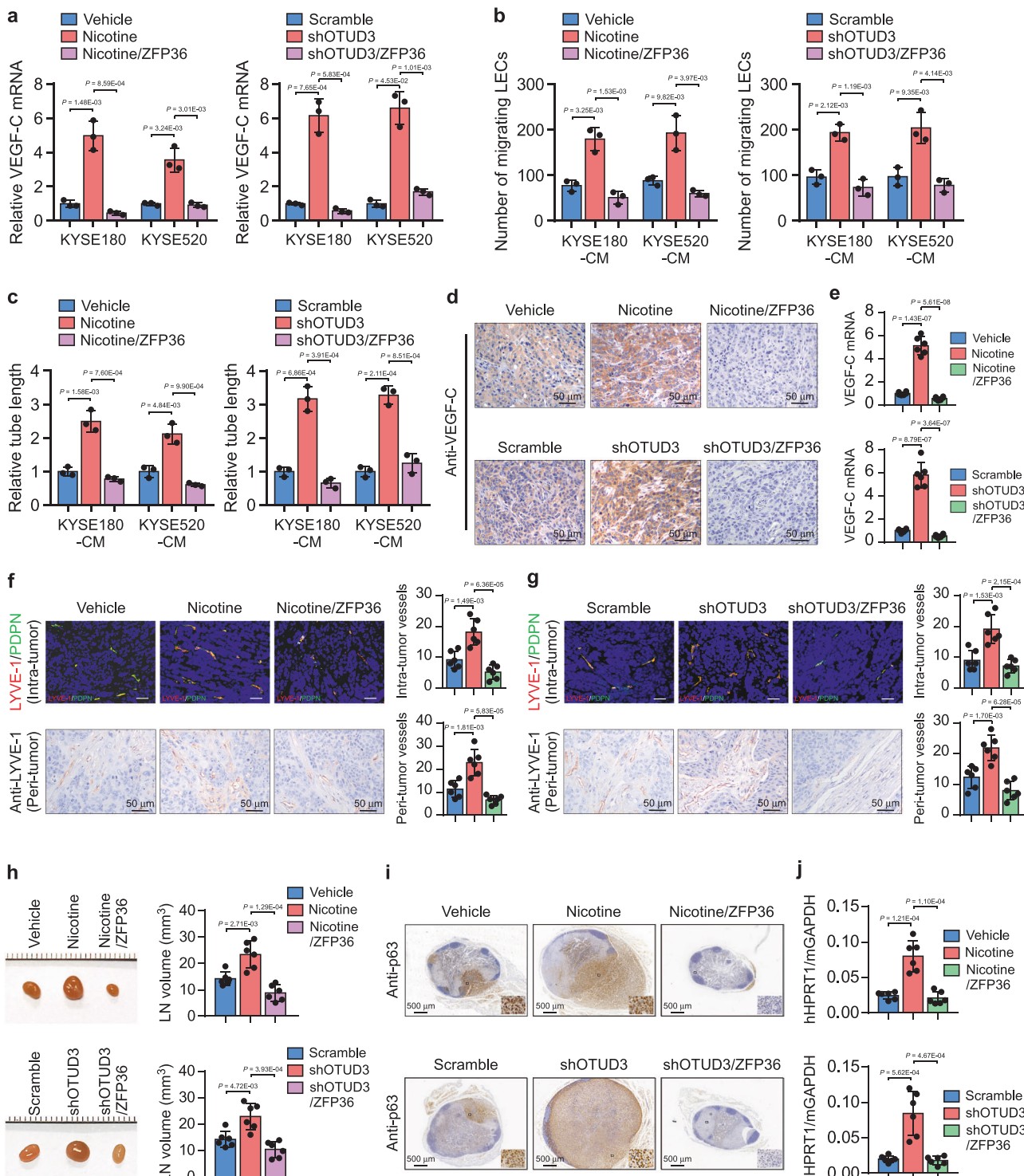

**Fig. 6 Induction of VEGF-C mRNA decay abrogates lymphatic metastasis. a** qRT-PCR analysis of VEGF-C mRNA expression in the indicated esophageal cancer cells. **b** Quantification of migrating LECs incubated with the indicated CM. **c** Relative tube length of LECs under treatment of the indicated CM. **d** The role of ZFP36 in the inhibition of lymphatic metastasis was examined by the inguinal lymph node metastasis model. Shown are the representative IHC staining of VEGF-C in footpad tumors from each group (*n* = 6). Scale bars: 50 μm. **e** qRT-PCR analysis of VEGF-C mRNA expression in the indicated footpad tumors (*n* = 6/group). **f**, **g** Representative images and quantification of intra-tumoral and peri-tumoral lymphatic vessels in footpad tumors as indicated by dual-IF of LYVE-1 and PDPN and IHC staining of LYVE-1. Scale bars: 50 μm. **h** Representative image of inguinal LNs. The volumes of LNs from each group were quantified. **i** Representative IHC staining images of squamous cell carcinoma marker p63 in LNs. The p63-positive cells indicated the metastatic KYSE180 cells in LNs and were enlarged in the insets. Scale bars: 500 μm. **j** qRT-PCR analysis of human HPRT1 relative to mouse GAPDH in the LNs from each group. The ratio indicated the proportion of metastatic cells. Each error bar in **a**–**c** represents the mean ± SD of three biological replicates. Data in **e**–**h**, **j** represent the mean ± SD derived from tumor mouse models (*n* = 6 mice/group). Two-sided Student's *t* test was used for all panels. Source data are provided as a Source data file.

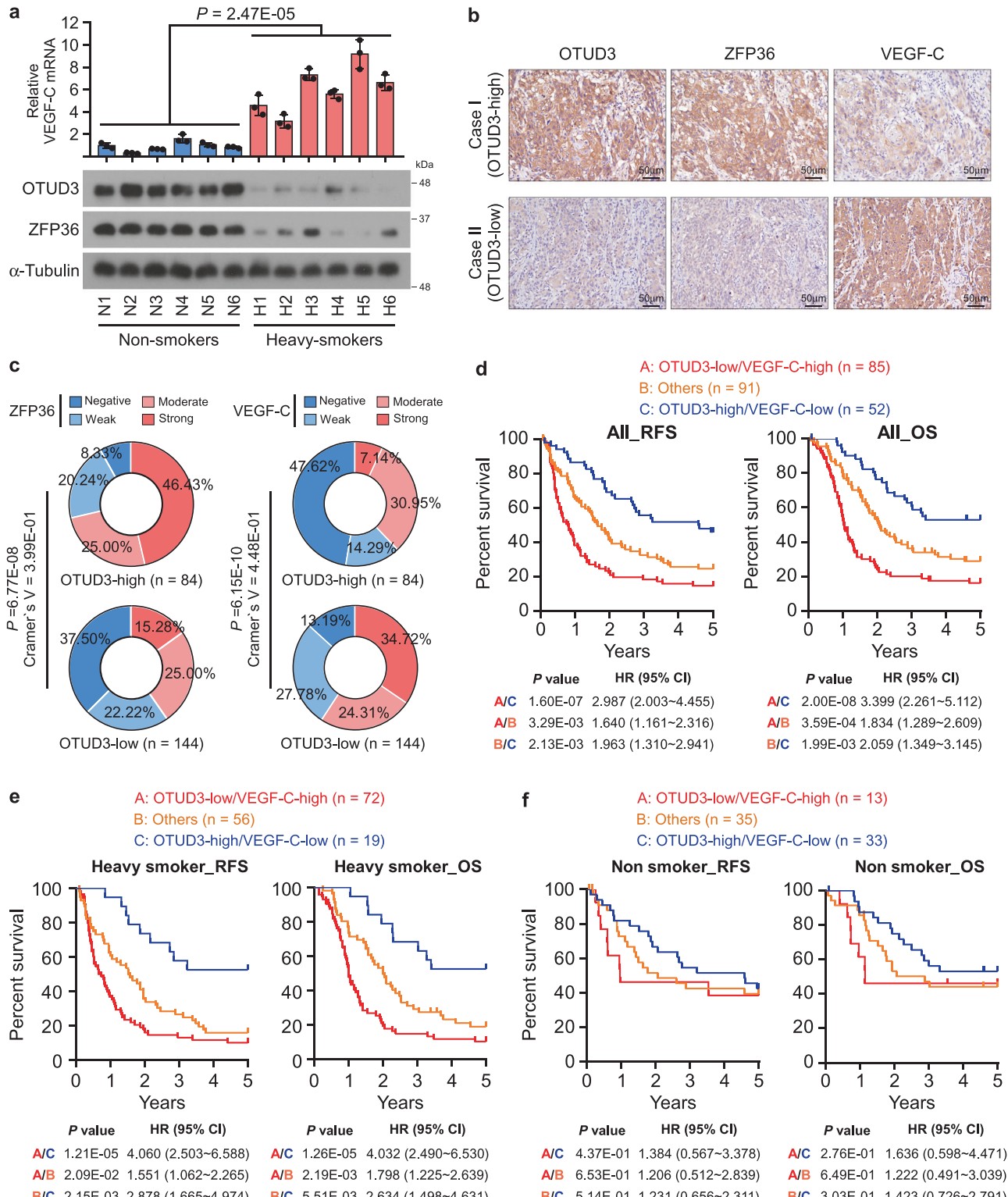

**Fig. 7 Clinical relevance of the OTUD3/ZFP36/VEGF-C axis. a** qRT-PCR of VEGF-C and western blot analysis of OTUD3 and ZFP36 in primary esophageal tumors derived from six non-smokers and six heavy smokers. Each error bar represents the mean ± SD of three biological replicates. Statistical significance was determined by one-way ANOVA compared with non-smokers. Immunoblots in this figure are representative of three biological replicates. **b** Representative images of OTUD3, ZFP36, and VEGF-C IHC staining in esophageal cancer patient specimens ($n = 228$). Scale bars: 50 μm. **c** Correlation analysis showed that OTUD3 was significantly associated with the expression scores of ZFP36 and VEGF-C. Two-sided $\chi^2$ test and Cramer's V were used to evaluate the correlation. **d–f** The patient specimens were divided into three groups, including OTUD3-low/VEGF-C-high, OTUD3-high/VEGF-C-low, and others. Kaplan–Meier survival curves and log-rank test were then used to test their significance in predicting RFS and OS of all (**d**), heavy-smoking (**e**), or non-smoking (**f**) esophageal cancer patients. The relative HRs between different signatures are indicated. Source data are provided as a Source data file.

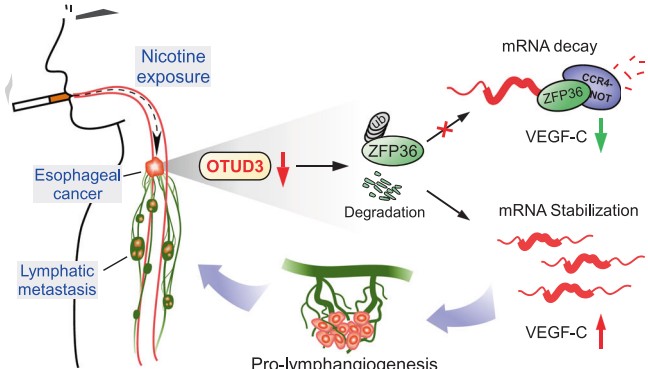

**Fig. 8 Study model.** Nicotine-mediated OTUD3 downregulation facilitates ZFP36 protein degradation and inhibits VEGF-C mRNA decay, leading to robust VEGF-C production, tumor-induced lymphangiogenesis, and lymphatic metastasis in esophageal cancer.

anti-ZFP36 (Merck Millipore, ABE285) or anti-CNOT1 (Bethyl, A305-787A) antibody and washed five times with the lysis buffer. The retrieved pellets were then subjected to RT-PCR analysis using the qRT-PCR primers of VEGF-C-3′UTR. ACTB was used as a negative control. Information on primers and oligonucleotides used in this study is provided as Supplementary Data 2.

**RNA synthesis, labeling, and pulldown**. RNA for in vitro experiments was transcribed with T7 or SP6 RNA polymerase (Ambion) from PCR-amplified templates with T7 forward 5′-TAATACGACTCACTATAG-3′ and SP6 Reverse 5′-ATT-TAGGTGACACTATAG-3′ primers. According to the manufacturer's instructions, the RNA fragments, including 5′UTR, CDS, 3′UTR, 3′UTR-mut, and 3′UTR-AS, were purified and biotin-labeled using the RNA 3′-End Biotinylation Kit (Thermo Fisher Scientific). The in vitro binding assays of biotin-labeled VEGF-C mRNA fragments and ZFP36 protein were performed. Briefly, cells ($1 \times 10^7$) were lysed with 1 ml of binding buffer (50 mM Tris-HCl pH 7.9, 10% glycerol, 100 mM KCl, 5 mM MgCl$_2$, 10 mM β-ME, 0.1% NP-40, 1 mM PMSF, 1× Superase-in, and 1× protease inhibitor cocktail). Labeled RNA (50 pmol) was incubated with cell lysates for 1 h at RT. Then 50 μl of washed streptavidin-conjugated magnetic beads (Thermo Fisher Scientific) were added to each reaction and incubated at RT for 30 min. Beads were washed five times, and the retrieved protein was subjected to western blot analysis.

**RNA-seq data**. KYSE180 cells were treated with DMSO or CSE for 24 h and then subjected to RNA extraction and RNA-seq analysis. Differential expression analysis was implemented using DESeq2 package. The RNA-seq data have been deposited in the National Center for Biotechnology Information Sequence Read Archive database (https://www.ncbi.nlm.nih.gov/sra/PRJNA678868).

**Statistical analysis**. Statistical analyses are performed using the SPSS version 19.0 statistical software package and Graph-Pad Prism 8 version 8.3.0 software (GraphPad software, La Jolla, CA, USA). Statistical tests for data analysis included log-rank test, $\chi^2$ test (two-sided), and Student's $t$ test (two-sided). Multivariate statistical analysis is performed using a Cox regression model. The strength of the relationship is evaluated by the Cramer's V correlation coefficient. $P < 0.05$ was considered statistically significant.

**Reporting summary**. Further information on research design is available in the Nature Research Reporting Summary linked to this article.

## Data availability
The RNA-seq data have been deposited in the National Center for Biotechnology Information Sequence Read Archive (SRA) database (https://www.ncbi.nlm.nih.gov/sra/PRJNA678868). The mass spectrometry proteomics data have been deposited to the ProteomeXchange Consortium via the iProX partner repository with the dataset identifier PXD028751. All the other data supporting the findings of this study are available within the article and its Supplementary Information files. A reporting summary for this article is available as a Supplementary Information file. Source data are provided with this paper.

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

## Acknowledgements

This work was supported by the National Key Research and Development Program of China (No. 2020YFA0509400), National Natural Science Foundation of China (Nos. 81672854, 81773106, 82072609, 81872383, 82173302, 81530082, and 91740118), Natural Science Foundation of Guangdong Province (Nos. 2017A030306019, 2018B030311060), Pearl River S&T Nova Program of Guangzhou (No. 201710010163), the Fundamental Research Funds for the Central Universities (19ykzd45), Guangdong Esophageal Cancer Institute Science and Technology Program (M201805), and the program for Excellent Talents in Cancer Centre (16zxyc01).

## Author contributions

Y.L., M.W., Y.X., and M.Y. carried out most of the experimental work; they collected and analyzed the data. Y.L., M.Y., Y.J., and D.S. conducted the RNA-seq, qRT-PCR, luciferase reporters, ChIP, and PLA assays. Y.L., M.Y., Y.X., J.C., Xin Chen, and L.K. collected tissues, patient information, and conducted IHC and survival analysis. M.W., Y.X., Xiangfu Chen, and X.H. conducted the western blot analysis, plasmid constructions, and IP assays. M.W., Y.L., J.C., and Xiangfu Chen conducted animal studies. Y.O., J.C., and J.B. conducted cell culture. M.Y. and Y.X. performed the in vitro studies. C.L. and L.S. raised the concept, design the experiments, wrote the manuscript, and supervised the project. All authors reviewed the manuscript.

## Competing interests

The authors declare no competing interests.
