## [Peer Review File · Nature Communications]

Nicotine-mediated OTUD3 downregulation inhibits VEGF-C mRNA decay to promote lymphatic metastasis of human esophageal cancerREVIEWER COMMENTS

Reviewer #1 (Expertise: LN metastasis, Remarks to the Author):

Wang et al. present a study of the function of OTUD3 in esophageal cancer and demonstrate an inverse relationship between OTUD3 and VEGF-C in studies in vitro. Molecular studies in vitro are strong and demonstrate mechanism of regulation of VEGF-C expression related to OTUD3. However, link between OTUD3, VEGF-C and poor outcome in vivo is not as well established. Because of many different roles of OTUD3, it is difficult to establish that the effects on tumor progression observed in vivo are because of VEGF-C modulation.

Specific comments:

1. Previous studies of OTUD3 are contradicting. In breast cancer, OTUD3 was reported to play a suppressive role in breast tumorigenesis through stabilizing PTEN protein (Yuan et al. *Nat Cell Biol.* 2015 Sep; 17(9):1169-81). In lung cancer, OTUD3 is highly expressed and its high expression correlates with poor survival (Du et al., *Nat Commun.* 2019 Jul 2; 10(1):2914.). The interpretation has been that OTUD3 could elicit tumor-suppressing or tumor-promoting activities depending on a cell- and tissue-dependent context. With this in mind, it would be important to understand in which context in esophageal cancer OTUD3 expression has protective role. In human esophageal cancer, what is the relationship between OTUD3 expression and cancer stage, VEGF-C expression and cancer stage? Authors indicate that 62% OTUD3 high cancers are VEGF-C high, still large number of cancers that are OTUD3 low (41%), are VEGF-C high. OTUD3/VEGF-C relationship in vivo is not that straightforward. This needs to be further addressed.
2. Fig. 2E, 6E: Increase of intratumoral lymphatic vessels is not convincing. Additional markers should be used to exclude that LYVE-1 signal is not on macrophages and make sure that the signal is derived from lymphatic endothelial cells (podoplanin, Prox1 can be used as additional lymphatic markers, or CD31 as an endothelial marker).
3. Fig. 3H: Please show RNA levels of VEGF-C when OTUD3 expression is modulated in addition to the VEGF-C immunostaining.
4. Fig. 6H: It is difficult to discern tumor cells in the lymph node in the nicotine group, as everything is light brown/yellow. Please show pattern of expression of p63 in primary tumors used in those experiments and provide better quality data for lymph nodes. Are all tumor cells labeled with p63 in primary tumors?
5. Please revisit literature citations to include some key original papers instead of many reviews. For example, ref. 16 and 17, one review could be eliminated. Instead, cite key original papers: Skobe M et al. Induction of tumor lymphangiogenesis by VEGF-C promotes breast cancer metastasis. *Nat Med.* 2001 Feb; 7(2): 1 and Stacker SA et al. VEGF-D promotes the metastatic spread of tumor cells via the lymphatics. *Nat Med.* 2001 Feb; 7(2):186-91.

Reviewer #2 (Expertise: Nicotine, cancer, Remarks to the Author):

In efforts to better understand cigarette-smoking induced lymphatic metastasis in human esophageal cancer, the authors investigate a specific deubiquitinase, OTUD3, which is modified by nicotine exposure, promoting lymphatic metastasis in-vivo. The authors demonstrate for the first time that this is mediated in part by ZFP36 and VEGF-C and suggest OTUD3 or ZFP36 as therapeutic targets for treatment of lymphatic metastasis of esophageal cancer. The broad use of human cell, mouse, and patient data in the study provides a strong rationale for the pathway elucidated. This article is beautifully written with a clear study model graphic, providing accessibility to all readers regardless of background.

- Regarding the patient studies with heavy smokers, were they still smoking at time of the tissue biopsy? Had any patients quit smoking previous to the biopsies (i.e. were any reformed smokers,

and for how many years)?

- Others have used direct (nose-only) cigarette exposure models for mice, replicative of firsthand smoking. Can you comment on how your results may differ?
- The nicotine administration was daily, but what was the length of exposure? The entire month? Reference 46 you refer to doesn't have this information in it, please check if this is the correct reference here.
- What day was the footpad tumor injection done? Day 1, same as first day of nicotine exposure? Or once the smoking-like microenvironment was created? Please clarify.
- There are no magnification bars or mention of the magnification for any of the immunohistochemical images.
- Figure 6D, are these representative images or was staining done for all footpad tumors? Was it quantified? Same question for 6H.
- While cigarette smoking is still prevalent in developing countries, electronic cigarettes are becoming increasingly popular in other areas, especially as an alternative to cigarette smoking, likely in efforts to quit smoking. Unfortunately, most e-cigarette liquids contain nicotine in varying concentrations mixed with other constituents that are unknown. Will your future studies address e-cigarettes, esophageal cancer and OTUD3 expression?

Reviewer #3 (Expertise: RBP, mRNA decay, Remarks to the Author):

In this study, Wang et al. demonstrate that nicotine and its derivatives downregulate the expression of the deubiquitinase OTUD3 in esophageal cancer cells on both the mRNA and protein level. The decrease in OTUD3 expression via transcriptional regulation involving the PI3K/FOXO signaling pathway is plausible and convincing. The authors further demonstrate that OTUD3 downregulation is clinically relevant since it is associated with a higher number of lymphatic vessels in esophageal tumors, increased lymph node metastasis and reduced survival of heavy-smoking patients. By using both in vitro cell culture and in vivo mouse models, the authors demonstrate that nicotine-induced lymphatic metastasis can be attenuated by ectopic OTUD3 expression, thereby establishing OTUD3 as important regulator of tumor-induced lymphangiogenesis. Since VEGF, well known for its role in pathological angiogenesis, is upregulated by nicotine in an OTUD3-dependent manner, the authors set out to explore the underlying molecular mechanism. They reveal a post-transcriptional regulatory mechanism by which loss of OTUD3 causes enhanced proteolysis of the RNA-binding protein ZFP36, leading to enhanced stability of VEGF-C mRNA. The authors further demonstrate that OTUD3 serves as a K48-specific deubiquitinase of ZFP36 and thereby enhances ZFP36 protein expression. This is the most exciting part of the manuscript and the experimental evidence provided is ample and convincing.

Although the binding of ZFP36 to AREs embedded within the 3'UTR of VEGF mRNA, its ZFP36-mediated decay as well as the antitumor and antiangiogenic function of ZFP36 have already been described, this manuscript uncovers a yet unknown mechanism by which nicotine causes ubiquitin-dependent proteolytic degradation of ZFP36. Since loss of OTUD3/ZFP36 expression is associated with elevated expression of VEGF-C mRNA in clinical specimens derived from heavy-smoking patients, the present study makes an important contribution towards a better understanding of smoking-induced esophageal tumors. My comments are minor and pertain to some control western blots as well as a more balanced discussion of the results.

Specific Comments

Figure 2A-C: The authors should show OTUD3 protein expression by western blot analysis in KYSE1280 and KYSE520 cells.

Figure 3E, F: Knock-down of ZFP36 should be assessed by western blot analysis.

Figure 4F: Why does nicotine treatment enhance ZFP36 ubiquitination in presence of transfected OTUD3, given that the expression level of OTUD3 does not change compared to transfected OTUD3 without nicotine treatment. This point should be discussed.

4G, last lane: I assume the plus in the middle should be a minus.

Figure 5A: The authors could explain better why VEGF-C, but not VEGF-D, is controlled by the OTUD3-ZFP36 axis. Is the 3'UTR of VEGF-C different from the VEGF-D 3'UTR? Does only VEGF-C mRNA contain an ARE? What about other VEGF isoforms? A scheme would probably help to understand the differences.

Figure 5B, F, J: For the RIP assays the authors should provide western blots to show enrichment of the immunoprecipitated proteins (i.e. ZFP36 and CNOT1). The amount of recovered VEGF mRNA in the ZFP36 RIP following nicotine treatment, ZFP36 knock-down or ZFP36 overexpression might simply reflect the differential expression of ZFP36 in these conditions. Therefore, assessment of recovered ZFP36 by western blot analysis would be informative, and the result should be discussed accordingly.

The authors propose that the OTUD3-ZFP36-VEGF axis is of central importance for LN metastasis, which is supported by previous work on the role of VEGF in lymphangiogenesis. However, the experimental evidence provided by the authors does not formally prove that VEGF is indeed the key target of ZFP36 for the effects on LN metastasis. Hence, the authors should phrase their model more carefully, and state that other targets of ZFP36 may also contribute to phenotypic differences. For instance, the inhibitory effect of ZFP36 on growth of LN metastasis may not only be due to suppression of VEGF-C expression, but also to the pro-apoptotic effect of ZFP36 in tumor cells (first shown by Blackwell, PMID: 10763822). This pertains to the interpretation of Figure 6G-I, and to the discussion in general.

The manuscript stands in a line of studies demonstrating a tumor-suppressive effect of ZFP36, both in animal models (e.g. PMID: 12789264) and in human cancer (e.g. PMID: 21875902, PMID: 19491267, PMID: 19697322). In fact, some of these studies have already pointed to VEGF as an important target of ZFP36 with respect to its tumor-suppressive effects. To my knowledge, the first report of VEGF as a target of ZFP36 was in PMID: 17855506. The authors should present the background literature on ZFP36 to place the current study in proper perspective.

page 2, second last line: decay of messenger RNA (mRNA)

page 11, line 6: loss of ZFP36 expression

Antibodies used for western blotting should be listed in the methods section.

Reviewer #4 (Expertise: Esophageal cancer, Remarks to the Author):

In this study, the authors evaluated the significance of nicotine-mediated OTUD3 regulation and lymphangiogenesis in esophageal cancer. Interestingly, the OTUD3 expression was significantly associated with smoking status in clinical esophageal cancer patients, and the authors clarified the mechanism of OTUD3-regulated lymphangiogenesis via regulation of the ZFP36/VEGF-C axis. This report is intriguing; however, the current study does not meet the publishing criteria in this journal. Therefore, I raised several points to improve the content of the report.

In this study, the authors evaluated the significance of nicotine-mediated OTUD3 regulation and lymphangiogenesis in esophageal cancer. Interestingly, the OTUD3 expression was significantly associated with smoking status in clinical esophageal cancer patients, and the authors clarified the mechanism of OTUD3-regulated lymphangiogenesis via regulation of the ZFP36/VEGF-C axis. This report is intriguing; however, the current study does not meet the publishing criteria in this journal. Therefore, I raised several points to improve the content of the report.

Query

1. The authors focused on the nicotine effect against esophageal cancer cells, and the expression of OTUD3 in esophageal cancer cells was induced by nicotine treatment. Interestingly, the relation of OTUD3 expression in resected samples and smoking status was significant, and the data might be very interesting. However, esophageal cancer patients might stop smoking just before the resection surgery to prevent postoperative pulmonary complications. Why was the OTUD3 alteration in resected samples kept on an operative day? The comparison between current smokers and ever smoker may be important.

How did the nicotine treatment regulate the PI3K/AKT/FOXO1 signaling for suppressing the OTUD3 expression in esophageal cancer?

The relation of the OTUD3 expression and never-smoker, ever-smoker, and current smoker just before diagnosis should be shown as data.

The prognostic value and expression significance of FOXO1, ZFP36, and VEGF-C in esophageal cancer patients with or without smoking history should be shown as data. Kaplan-Meyer curves and clinicopathological tables should be used to show them. If the expressions were important for VEGF-C-related angiogenesis in esophageal patients with smoking history, the expression significance in non-smoker might be weak same as OTUD3.

In this study, the authors clarified the interesting relationship between nicotine, PI3K/AKT/FOXO1, OTUD3, ZFP36, and VEGFC in esophageal cancer. However, the story might be complex. Moreover, the relation should be validated in clinical esophageal cancer samples statistically.

IRB-approved number should be described in the materials and methods section.

Antibody information of p63 for IHC and GAPDH and tubulin for WB should be described in the materials and method section. Why did the authors change the antibodies of OTUD3 for IHC and WB

The information on adjuvant therapy in esophageal cancer patients should be described in the materials and method section.

Reviewer #5 (Expertise: DUBs, cancer, Remarks to the Author):

The authors investigated the link between nicotine addiction and esophageal tumorigenesis, notably lymphatic metastasis of esophageal cancer cells. The authors found that the deubiquitinase OTU domain-containing protein 3 (OTUD3) is downregulated by nicotine administration and its expression is correlated with poor prognosis in heavy-smoking esophageal cancer patients. The results support a model whereby nicotine-mediated OTUD3 downregulation is associated with tumor-induced lymphangiogenesis. Mechanistically, they found that OTUD3 directly

interacts with ZFP36 ring finger protein (ZFP36) resulting in its stabilization. This effect is associated with an inhibition of FBXW7-mediated K48-linked polyubiquitination. Next, the authors established that ZFP36 binds the mRNA of VEGF-C at its 3'-UTR and to promote its degradation. The authors proposed that nicotine induces downregulation of OTUD3 which leads to ZFP36 downregulation, which in turn promotes VEGF-C production and lymphatic metastasis in esophageal cancer.

Overall, this is a very interesting study that links nicotine addiction to OTUD3 and VEGF-C signaling and their impact on lymphatic metastasis. The OTUD3-VEGF-C signaling might be potentially exploited for the treatment of human esophageal cancer. The study is quite novel, and the data are in general convincing. The statistics appear adequate. I think this study could be of interest to a wide audience of researchers and clinicians. Nonetheless, I think the following comments need to be addressed before consideration of the manuscript for publication.

1) In Figure 1B, immunoblotting for OTUD3 protein in the various cell types treated with cigarette smoke cigarette need to be shown to corroborate the mRNA expression data.

2) Figure 2, The authors need to be explicit in their figures and legend to indicate whether they conducted overexpression or RNAi knockdown.

3) In general, the authors need to better describe the figures in the legends. In addition, the authors need to indicate the number of repetitions for all figures, e.g., western blots and RNA pull downs.

4) Figure 4. As described in material and methods, the IP-immunoblotting was not conducted under denaturing conditions. The authors could not exclude the possibility that the ubiquitination signals observed by immunoblotting might be contributed by additional ZFP36-associated proteins. Thus, the authors should repeat some of these immunoprecipitation experiments by first denaturing the extract in 1% SDS in conjunction with heat inactivation and then dilute the samples in an adequate buffer before conducting the IP. This will help excluding the possibility that other proteins are ubiquitinated.

5) The authors have the tendency to overstate the conclusions of the manuscript. For instance, the authors state "Nicotine-mediated OTUD3 downregulation promoted lymphatic metastasis by inducing substantial tumor-induced lymphangiogenesis".

But the authors did not carefully demonstrate that modulation of OTUD3, induces tumor-induced lymphangiogenesis. The results shown by the authors in figure 2 are quite preliminary.

6) The authors need to repeat some of the key experiments on the biological impact of OTUD3 with a catalytic dead mutant to further corroborate findings on substrate deubiquitination.

7) The authors consider the E3 ligase FBXW7 for targeting the ubiquitination of ZFP36. The authors could consider conducting shRNA knockdown of this ubiquitin ligase and determine the impact on ZFP36 stability and VEGF-C signaling. For example, does FBXW7 depletion rescues the effects of OTUD3 depletion. In addition, FBXW7 is known as a tumor suppressor, and it should be discussed how this tumor suppressor activity could be integrated within the OTUD3-ZFP36-VEGF-C signaling axis.

8) It might be interesting to determine by confocal microscopy, the subcellular localization of OTUD3 and ZFP36 by immunostaining to determine whether these factors have similar subcellular compartmentalization.

9) The mass spectrometry data obtained through the purification of OTUD3-associated proteins should identify several peptides for each protein. The authors need to provide the data for all peptides counts and deposited in a resource database.

10) The authors show the effects of knockdowns, sometimes with two shRNAs, sometimes with one shRNA only. It should be two shRNAs, ideally throughout the manuscript. This will ensure that the observed effects are not due to off-target effects. In addition, the authors indicate shRNA and siRNA, this has to be clarified. If siRNAs are used, two different siRNAs should be used to exclude

potential off target effects. In the shRNA experiments, what is control? Non-transfected, empty vector, or a scrambled sequence? This also needs to be indicated.

11) The discussion should be broadened to put everything in context and to cover the known effect and mechanisms of action of nicotine.

Other comments

12) List the source and catalogues numbers of all antibodies used, e.g., anti-p-Akt, anti-Akt, anti-p-ERK1/2, anti- ERK1/2, anti-p-p38, anti-p38 antibodies, Ubiquitin antibody, etc.

13) The histograms need to be uniform. Sometimes histograms are shown with the error bar only (Ex: Fig 1), but all points are shown with the error bar in figure 6G, I. Include error bar with individual points in all histograms.

14) Include scale barre in all IHC pictures (at least one picture per panel).

15) In the introduction, the authors state Deubiquiyinase are pleiotropic players... This might need to be reformulated

16) Correct all typos, eg., Ployubiquitination analysis.

17) Perhaps consider changing the tense of some sentences. In material and methods e.g., Details of the IHC method are (instated of were) provided in the supplementary information. In the abstract, Here we show (instead of Here we showed). At the mechanistic level, OTUD3 directly interacts (instead of interacted) with ZFP36 ring finger protein (ZFP36) and stabilized it by inhibiting FBXW7-mediated K48-linked polyubiquitination.

18) Manuscript to format according to the style of Nature Comm.

Response to reviewers' comments and suggestions:

Reviewers' comments:

Reviewer #1 (Expertise: LN metastasis, Remarks to the Author):

Wang et al. present a study of the function of OTUD3 in esophageal cancer and demonstrate an inverse relationship between OTUD3 and VEGF-C in studies in vitro. Molecular studies in vitro are strong and demonstrate mechanism of regulation of VEGF-C expression related to OTUD3. However, link between OTUD3, VEGF-C and poor outcome in vivo is not as well established. Because of many different roles of OTUD3, it is difficult to establish that the effects on tumor progression observed in vivo are because of VEGF-C modulation.

Specific comments:

1. Previous studies of OTUD3 are contradicting. In breast cancer, OTUD3 was reported to play a suppressive role in breast tumorigenesis through stabilizing PTEN protein (Yuan et al. *Nat Cell Biol.* 2015 Sep;17(9):1169-81). In lung cancer, OTUD3 is highly expressed and its high expression correlates with poor survival (Du et al., *Nat Commun.* 2019 Jul 2;10(1):2914.). The interpretation has been that OTUD3 could elicit tumor-suppressing or tumor-promoting activities depending on a cell- and tissue-dependent context. With this in mind, it would be important to understand in which context in esophageal cancer OTUD3 expression has protective role. In human esophageal cancer, what is the relationship between OTUD3 expression and cancer stage, VEGF-C expression and cancer stage? Authors indicate that 62% OTUD3 high cancers are VEGF-C high, still large number of cancers that are OTUD3 low (41%), are VEGF-C high. OTUD3/VEGF-C relationship in vivo is not that straightforward. This needs to be further addressed.

Response: We thank the reviewer for the comment. Indeed, OTUD3 was reported to play a context-dependent role in cancers. It suppresses tumorigenesis in breast, colon, liver, brain, and cervical cancer by stabilizing PTEN and p53, while promoting lung tumorigenesis by stabilizing GRP78 [1-4]. For esophageal cancer, cigarette smoking is recognized as a significant prognostic factor, contributing to tumor malignant progression [5]. In this study, we found that OTUD3 was highly expressed in normal esophageal epithelial cells (NEECs, Fig. 1b) and normal esophageal tissues (Fig. 1d). Notably, OTUD3 was substantially decreased under nicotine exposure in esophageal cancer cells and heavy-smoking patient specimens (Fig. 1a-d). OTUD3 stabilized the tumor-suppressive ZFP36 protein and induced rapid mRNA decay of VEGF-C. Downregulation of OTUD3 promoted tumor-induced lymphangiogenesis and was essential for nicotine-induced lymphatic metastasis, leading to earlier tumor relapse and shorter patient survival, especially in heavy smokers. These observations further suggest that OTUD3 plays a tumor-suppressive role in esophageal cancer. This point has been further discussed in the revised manuscript.

Further analysis of the IHC data indicated that OTUD3 expression negatively correlated with cancer stages (χ^2 test, $P < 0.001$, Cramer's V = 0.612; Supplementary Fig. 2a and Supplementary Table 2), while VEGF-C expression was positively associated with cancer stages (χ^2 test, $P < 0.001$, Cramer's V = 0.403; Supplementary Fig. 8a and Supplementary Table 5). These data have been incorporated into the revised manuscript.

In the initial Figure 7B, we showed that ~62% OTUD3 high cancers are VEGF-C low, while a relatively high portion of cancers that are OTUD3 low (~41%), are VEGF-C low. The relationship seems not that straightforward. To address this concern, we first analyzed the strength of the relationship. Notably, the Cramer's V correlation coefficient between OTUD3 and VEGF-C expression in the initial Figure 7B was 0.202, indicating a strong (defined as > 0.15) [6] relationship between OTUD3 and VEGF-C expression. However, this figure might not reflect the correlation very well, probably because the IHC scores were grouped into high or low expression categories. We then assessed the correlation between OTUD3 and VEGF-C expression scores. As shown in Fig. 7c, the proportions of strong VEGF-C expression were robustly increased in OTUD3-low specimens; while the proportions of negative VEGF-C expression were decreased in the OTUD3-low group. Notably, the Cramer's V correlation coefficient of the test was 0.448, indicating a very strong (defined as > 0.25) [6] relationship between these two variables (Fig. 7c). Since this new analysis might better represent that OTUD3 regulates VEGF-C expression levels in esophageal cancer, we have replaced the figure along with the Cramer's V correlation coefficient (Fig. 7c). Importantly, the Kaplan-Meier survival estimates showed that patients with combined low OTUD3 and high VEGF-C expression suffered the worse RFS and OS, especially in the heavy-smoking patients, revealing that the OTUD3/VEGF-C axis was indeed associated with poor clinical outcomes of esophageal cancer (Fig. 7d-f).

Reference:

1. Yuan L, et al. Deubiquitylase OTUD3 regulates PTEN stability and suppresses tumorigenesis. *Nature cell biology* 17, 1169-1181 (2015).
2. Pu Q, Lv YR, Dong K, Geng WW, Gao HD. Tumor suppressor OTUD3 induces growth inhibition and apoptosis by directly deubiquitinating and stabilizing p53 in invasive breast carcinoma cells. *BMC cancer* 20, 583 (2020).
3. Du T, et al. The deubiquitylase OTUD3 stabilizes GRP78 and promotes lung tumorigenesis. *Nature communications* 10, 2914 (2019).
4. Liu YZ, Du XX, Zhao QQ, Jiao Q, Jiang H. The expression change of OTUD3-PTEN signaling axis in glioma cells. *Annals of translational medicine* 8, 490 (2020).
5. Kuang, J. J. et al. Smoking Exposure and Survival of Patients with Esophagus Cancer: A Systematic Review and Meta-Analysis. *Gastroenterol Res Pract.* 2016;2016:7682387.
6. Haldun Akoglu. User's guide to correlation coefficients. *Turk J Emerg Med.* 2018 Aug 7;18(3):91-93.

2. Fig. 2E, 6E: Increase of intratumoral lymphatic vessels is not convincing. Additional markers should be used to exclude that LYVE-1 signal is not on macrophages and make sure that the signal is derived from lymphatic endothelial cells (podoplanin, Prox1 can be used as additional lymphatic markers, or CD31 as an endothelial marker).

Response: We appreciate the comment and this point is well taken. As suggested by the reviewer, we further performed dual-immunofluorescence (IF) staining of lymphatic endothelial cell markers LYVE-1 and podoplanin (PDPN) as previously reported [1-2], to examine the intratumoral lymphatic vessels. Notably, the dual-IF staining indicated similar results as the original findings, showing that nicotine increased the number of intratumoral lymphatic vessels, while overexpression of OTUD3 or ZFP36 robustly reduced the number of vessels (Fig. 2e, Fig. 6f, g, and Supplementary Fig. 7b, c). These results have been incorporated into the revised manuscript.

Reference:

1. Sina Tadayon, et al. Lymphatic Endothelial Cell Activation and Dendritic Cell Transmigration Is Modified by Genetic Deletion of Clever-1. *Front Immunol.* 2021 Mar 4;12:602122.
2. Kai Song, et al. Lenalidomide Inhibits Lymphangiogenesis in Preclinical Models of Mantle Cell Lymphoma. *Cancer Res.* 2013 Dec 15;73(24):7254-64.

3. Fig. 3H: Please show RNA levels of VEGF-C when OTUD3 expression is modulated in addition to the VEGF-C immunostaining.

Response: This suggestion has been well taken. As shown in Supplementary Fig. 4e, the qRT-PCR analysis indicated that nicotine administration increased VEGF-C mRNA expression in subcutaneous KYSE180 tumors, while ectopic expression of OTUD3 abrogated this effect to retain VEGF-C mRNA at low levels. This result has been incorporated into the revised manuscript.

4. Fig. 6H: It is difficult to discern tumor cells in the lymph node in the nicotine group, as everything is light brown/yellow. Please show pattern of expression of p63 in primary tumors used in those experiments and provide better quality data for lymph nodes. Are all tumor cells labeled with p63 in primary tumors?

Response: These point has been well taken. As shown in Supplementary Fig. 7d-e, the IHC staining indicated that all tumor cells highly expressed the squamous cell carcinoma marker p63 in primary KYSE180 tumors. The p63-positive cells in lymph nodes indicated the metastatic KYSE180 cells showing that nicotine and OTUD3 downregulation robustly promoted lymph node metastasis. Better-quality data for lymph nodes have been provided in the revised manuscript (Fig. 6i).

5. Please revisit literature citations to include some key original papers instead of many reviews. For example, ref. 16 and 17, one review could be eliminated. Instead, cite key original papers: Skobe M et al. Induction of tumor lymphangiogenesis by VEGF-C promotes breast cancer metastasis. *Nat Med.* 2001 Feb;7(2):1 and Stacker SA et al. VEGF-D promotes the metastatic spread of tumor cells via the lymphatics. *Nat Med.* 2001 Feb;7(2):186-91.

Response: This point has been well-taken and the reference has been modified.

Reviewer #2 (Expertise: Nicotine, cancer, Remarks to the Author):

In efforts to better understand cigarette-smoking induced lymphatic metastasis in human esophageal cancer, the authors investigate a specific deubiquitinase, OTUD3, which is modified by nicotine exposure, promoting lymphatic metastasis in-vivo. The authors demonstrate for the first time that this is mediated in part by ZFP36 and VEGF-C and suggest OTUD3 or ZFP36 as therapeutic targets for treatment of lymphatic metastasis of esophageal cancer. The broad use of human cell, mouse, and patient data in the study provides a strong rationale for the pathway elucidated. This article is beautifully written with a clear study model graphic, providing accessibility to all readers regardless of background.

- Regarding the patient studies with heavy smokers, were they still smoking at time of the tissue biopsy? Had any patients quit smoking previous to the biopsies (i.e. were any reformed smokers, and for how many years)?

Response: We do appreciate the reviewer's comment. Indeed, smoking patients were strongly demanded to stop smoking before the resection surgery to prevent postoperative pulmonary complications. In this study, we focused on the heavy-smoking patients who were addicted to cigarettes and had an overtime smoking history estimated by pack-years no less than 20 ($PY \geq 20$). To address the reviewer's concern, we further divided the heavy-smoking patients were divided into the former and current categories as previously reported [1], in which former smokers were defined as those who quit smoking at least 1 year before surgery. Using this method, there were 108 current smokers and 39 former smokers in our patient cohort. Notably, a strong association was found between OTUD3 expression and smoking behavior (χ^2 test, $P < 0.001$, Cramer's V = 0.381), suggesting that OTUD3 expression was downregulated in the current smokers and could be recovered by quitting smoking (Supplementary Fig. 2b). These findings further indicate that OTUD3 expression is regulated by smoking in esophageal cancer.

Reference:

1. Kohei Shitara, et al. Heavy smoking history interacts with chemoradiotherapy for esophageal cancer prognosis: a retrospective study. *Cancer Sci.* 2010 Apr;101(4):1001-6.

- Others have used direct (nose-only) cigarette exposure models for mice, replicative of firsthand smoking. Can you comment on how your results may differ?

Response: We do appreciate the reviewer's comment and interest in this point. In this study, using *in vitro* chemical treatment, we identified that the major constituents of cigarette smoke, nicotine and its derivatives, but not Acetaldehyde or 4-Aminobiphenyl, induced robust OTUD3 downregulation in esophageal cancer cells. Moreover, via *in vivo* nicotine injection, we found that ectopic expression of OTUD3 abrogated nicotine-induced lymphatic metastasis. Interestingly, researchers have

established a direct (nose-only) cigarette exposure model for mice, which was supposed to better mimic smoking behavior, to explore smoking effects on diseases. Notably, this method is much more complicated with diverse types of hazard factors and stresses, which often induce alterations in the tissue microenvironment [1-4]. Thus, it would be interesting to use this model to investigate whether OTUD3 is downregulated, contributing to smoking-induced microenvironment changes and disease progression, which is on our future work plan.

Reference:

1. Jef Serré, et al. Enhanced lung inflammatory response in whole-body compared to nose-only cigarette smoke-exposed mice. *Respir Res.* 2021 Mar 17;22(1):86.
2. Shu J, et al. Comparison and evaluation of two different methods to establish the cigarette smoke exposure mouse model of COPD. *Sci Rep.* 2017 Nov 13;7(1):15454.
3. Manuela Rinaldi, et al. Long-term nose-only cigarette smoke exposure induces emphysema and mild skeletal muscle dysfunction in mice. *Dis Model Mech.* 2012 May;5(3):333-41.
4. Abderrahim Nemmar, et al. Waterpipe Tobacco Smoke Inhalation Triggers Thrombogenicity, Cardiac Inflammation and Oxidative Stress in Mice: Effects of Flavouring. *Int J Mol Sci.* 2020 Feb 14;21(4):1291.

• The nicotine administration was daily, but what was the length of exposure? The entire month? Reference 46 you refer to doesn't have this information in it, please check if this is the correct reference here.

Response: We are sorry that we did not describe this method clearly in the initial manuscript. For nicotine administration *in vivo*, nicotine (0.75 mg/kg, s.c., daily) [1] was subcutaneously injected with a 4 mm needle in mice one day before footpad tumor injection and last for an entire month. The success of nicotine injection *in vivo* was validated by the plasma cotinine (the nicotine metabolite) levels using ELISA assay at days 0, 10, 20, and 30 (Supplementary Fig. 3h). The reference was about the subcutaneous injection method and dose used for nicotine treatment *in vivo*, and has been moved to a more appropriate position. Appropriate modifications have been made in the METHODS section.

Reference:

1. Lefever TW, Thomas BF, Kovach AL, Snyder RW, Wiley JL. Route of administration effects on nicotine discrimination in female and male mice. *Drug Alcohol Depend* 204, 107504 (2019).

• What day was the footpad tumor injection done? Day 1, same as first day of nicotine exposure? Or once the smoking-like microenvironment was created? Please clarify.

Response: We are sorry that we did not describe this method clearly. Footpad tumor injection was done one day after nicotine injection and was defined as day 1. Considering that only nicotine was used for treatment, the description of the smoking-like microenvironment is not proper and has been deleted. Appropriate modifications have been in the METHODS section and the model scheme.

• There are no magnification bars or mention of the magnification for any of the immunohistochemical images.

Response: This has been corrected. Scale bars have been added in immunohistochemical images.

• Figure 6D, are these representative images or was staining done for all footpad tumors? Was it quantified? Same question for 6H.

Response: We are sorry that we did not state this point clearly. For Fig. 6d, IHC staining was done for all footpad tumors and shown with representative images from each group. To better quantify the levels of VEGF-C, we further examined the VEGF-C mRNA levels in these tumors using qRT-PCR analysis. As shown in Fig. 6e, VEGF-C mRNA was increased by nicotine or OTUD3 depletion in tumors, and these effects were abrogated by ZFP36 overexpression. This result has been incorporated into the revised manuscript.

Similarly, for Fig. 6i (the initial Fig. 6H), p63 staining was performed in all LNs and representative images from each group were shown. The images showed that ZFP36 expression substantially impaired nicotine- or shOTUD3-induced LN metastasis (Fig. 6i). Moreover, the proportion of metastatic KYSE180 in LNs was quantified by qRT-PCR analysis of human HPRT1 relative to mouse GAPDH as previously reported [1] (Fig. 6j). The relative mRNA ratio indicated that ZFP36 strikingly decreased the proportion of colonized tumor cells in LNs (Fig. 2i). Appropriate modifications have been made in the revised manuscript.

Reference:

1. Yu, F. et al. let-7 regulates self renewal and tumorigenicity of breast cancer cells. Cell 2007, 131, 1109-1123.

• While cigarette smoking is still prevalent in developing countries, electronic cigarettes are becoming increasingly popular in other areas, especially as an alternative to cigarette smoking, likely in efforts to quit smoking. Unfortunately, most e-cigarette liquids contain nicotine in varying concentrations mixed with other constituents that are unknown. Will your future studies address e-cigarettes, esophageal cancer and OTUD3 expression?

Response: We do appreciate the reviewer's comment and interest in this point. As stated by the reviewer, e-cigarettes are becoming popular in many areas as an alternative to cigarette smoking [1]. Studies are revealing that e-cigarettes can also do harm to human health and cause diseases, as vaping heats up various flavorings, nicotine, marijuana, or other potentially harmful substances [2-4]. Although there's no conclusive evidence that vaping causes esophageal cancer, certain studies suggest that vaping could potentially increase nicotine-related carcinogens *in vivo* [5-7]. Therefore, it would be of great significance to investigate whether OTUD3 is downregulated and play a critical role in e-cigarette-using esophageal cancer patients, which is on our future study plan. This point has been further discussed in the revised manuscript.

Reference:

1. Dai H, Leventhal AM. Prevalence of e-Cigarette Use Among Adults in the United States, 2014-2018. Jama 322, 1824-1827 (2019).
2. Choi K, Wills TA, Inoue-Choi M. E-cigarettes for smoking reduction: a piece of the public health puzzle.

Lancet Respir Med 9, 804-805 (2021).

3. Stokes AC, et al. Association of Cigarette and Electronic Cigarette Use Patterns With Levels of Inflammatory and Oxidative Stress Biomarkers Among US Adults: Population Assessment of Tobacco and Health Study. *Circulation* 143, 869-871 (2021).

4. Sultan AS, Jessri M, Farah CS. Electronic nicotine delivery systems: Oral health implications and oral cancer risk. *J Oral Pathol Med* 50, 316-322 (2021).

5. Hyun-Ji Kim, Ho-Sang Shin. Determination of tobacco-specific nitrosamines in replacement liquids of electronic cigarettes by liquid chromatography-tandem mass spectrometry. *J Chromatogr A*. 24;1291:48-55 (2013).

6. Bustamante G, et al. Presence of the Carcinogen N'-Nitrosornicotine in Saliva of E-cigarette Users. *Chem Res Toxicol* 31, 731-738 (2018).

7. Hiler M, et al. Electronic cigarette user plasma nicotine concentration, puff topography, heart rate, and subjective effects: Influence of liquid nicotine concentration and user experience. *Exp Clin Psychopharmacol* 25, 380-392 (2017).

Reviewer #3 (Expertise: RBP, mRNA decay, Remarks to the Author):

In this study, Wang et al. demonstrate that nicotine and its derivatives downregulate the expression of the deubiquitinase OTUD3 in esophageal cancer cells on both the mRNA and protein level. The decrease in OTUD3 expression via transcriptional regulation involving the PI3K/FOXO signaling pathway is plausible and convincing. The authors further demonstrate that OTUD3 downregulation is clinically relevant since it is associated with a higher number of lymphatic vessels in esophageal tumors, increased lymph node metastasis and reduced survival of heavy-smoking patients. By using both in vitro cell culture and in vivo mouse models, the authors demonstrate that nicotine-induced lymphatic metastasis can be attenuated by ectopic OTUD3 expression, thereby establishing OTUD3 as important regulator of tumor-induced lymphangiogenesis. Since VEGF, well known for its role in pathological angiogenesis, is upregulated by nicotine in an OTUD3-dependnet manner, the authors set out to explore the underlying molecular mechanism. They reveal a post-transcriptional regulatory mechanism by which loss of OTUD3 causes enhanced proteolysis of the RNA-binding protein ZFP36, leading to enhanced stability of VEGF-C mRNA. The authors further demonstrate that OTUD3 serves as a K48-specific deubiquitinase of ZFP36 and thereby enhances ZFP36 protein expression. This is the most exciting part of the manuscript and the experimental evidence provided is ample and convincing.

Although the binding of ZFP36 to AREs embedded within the 3'UTR of VEGF mRNA, its ZFP36-mediated decay as well as the antitumor and antiangiogenic function of ZFP36 have already been described, this manuscript uncovers a yet unknown mechanism by which nicotine causes ubiquitin-dependent proteolytic degradation of ZFP36. Since loss of OTUD3/ZFP36 expression is associated with elevated expression of VEGF-C mRNA in clinical specimens derived from heavy-smoking patients, the present study makes an important contribution towards a better understanding of smoking-induced esophageal tumors. My comments are minor and pertain to some control western blots as well as a more balanced discussion of the results.

Specific Comments

Figure 2A-C: The authors should show OTUD3 protein expression by western blot analysis in KYSE180 and KYSE520 cells.

Response: This point is well taken. OTUD3 protein expression in indicated KYSE180 and KYSE520 cells have been incorporated (Supplementary Fig. 3d).

Figure 3E, F: Knock-down of ZFP36 should be assessed by western blot analysis.

Response: This point has been well taken. Western blot analysis of ZFP36 has been provided (Fig. 3e, f).

Figure 4F: Why does nicotine treatment enhance ZFP36 ubiquitination in presence of transfected OTUD3, given that the expression level of OTUD3 does not change compared to transfected OTUD3 without nicotine treatment. This point should be discussed.

Response: We thank the reviewer for the comment. Interestingly, nicotine was found to inhibit the activity of proteasomes [1-2], suggesting that nicotine may generally inhibit protein degradation post-ubiquitin modifications, which might explain that nicotine could induce an increase in ZFP36 ubiquitination in presence of transfected OTUD3. Nevertheless, nicotine might also regulate other effectors for ZFP36 ubiquitination, which remains to be further explored. Interpretation of the result has been modified and this point has been discussed accordingly.

Reference:

1. Khosrow Rezvani, et al. Nicotine regulates multiple synaptic proteins by inhibiting proteasomal activity. *J Neurosci.* 2007 Sep 26;27(39):10508-19.
2. Sabine H van Rijt, et al. Acute cigarette smoke exposure impairs proteasome function in the lung. *Am J Physiol Lung Cell Mol Physiol.* 2012 Nov 1;303(9):L814-23.

4G, last lane: I assume the plus in the middle should be a minus.

Response: We thank the reviewer for pointing out this mistake. It has been corrected.

Figure 5A: The authors could explain better why VEGF-C, but not VEGF-D, is controlled by the OTUD3-ZFP36 axis. Is the 3'UTR of VEGF-C different from the VEGF-D 3'UTR? Does only VEGF-C mRNA contain an ARE? What about other VEGF isoforms? A scheme would probably help to understand the differences.

Response: We thank the reviewer for the comment. AREs are present in 5–8% of human genes [1]. The 3'UTRs of VEGF family genes are different (Supplementary Fig. 6a). Notably, except for VEGF-B, VEGF-A, VEGF-C and VEGF-D all contained AREs (Figure 5a and Supplementary Fig. 6a), suggesting their mRNA levels might be regulated via decay. Indeed, VEGF-A was reported as a direct target of ZFP36 in angiogenesis (blood vessels) [2]; however, the mRNA decay regulation of VEGF-C and VEGF-D in lymphangiogenesis (lymphatic vessels) remains unclear.

In this study, we found that the OTUD3-ZFP36 axis specifically induced the mRNA decay of VEGF-C, but not VEGF-D, through the conserved ARE to inhibit lymphangiogenesis. Notably, the specificity of mRNA decay is determined both by cis-acting ARE sequence and trans-acting RBPs including ZFP36, AUF-1, KSRP, HuR, TIA-1, and TIAR [3]. We speculated that the mRNA decay of VEGF-D might be regulated by other RBPs, which remains to be identified in the future. These descriptions have been incorporated into the revised manuscript.

Reference:

1. Bakheet T, Williams BR, Khabar KS. ARED 3.0: the large and diverse AU-rich transcriptome. *Nucleic acids research* 34, D111-114 (2006).

2. Essafi-Benkhadir K, Onesto C, Stebe E, Moroni C, Pages G. Tristetraprolin inhibits Ras-dependent tumor vascularization by inducing vascular endothelial growth factor mRNA degradation. *Molecular biology of the cell* 18, 4648-4658 (2007).
3. Garcia-Maurino SM, et al. RNA Binding Protein Regulation and Cross-Talk in the Control of AU-rich mRNA Fate. *Front Mol Biosci* 4, 71 (2017).

Figure 5B, F, J: For the RIP assays the authors should provide western blots to show enrichment of the immunoprecipitated proteins (i.e. ZFP36 and CNOT1). The amount of recovered VEGF mRNA in the ZFP36 RIP following nicotine treatment, ZFP36 knock-down or ZFP36 overexpression might simply reflect the differential expression of ZFP36 in these conditions. Therefore, assessment of recovered ZFP36 by western blot analysis would be informative, and the result should be discussed accordingly.

Response: We thank the reviewer for this comment. As recommended, the immunoprecipitated ZFP36 and CNOT1 proteins were examined in indicated RIP assays (Fig. 5f, j and Supplementary Fig. 6b). Notably, the amount of retrieved VEGF-C mRNA was related to recovered ZFP36 protein in the RIP assays, further suggesting that ZFP36 binds with the VEGF-C mRNA. These results have been incorporated and discussed accordingly.

The authors propose that the OTUD3-ZFP36-VEGF axis is of central importance for LN metastasis, which is supported by previous work on the role of VEGF in lymphangiogenesis. However, the experimental evidence provided by the authors does not formally prove that VEGF is indeed the key target of ZFP36 for the effects on LN metastasis. Hence, the authors should phrase their model more carefully, and state that other targets of ZFP36 may also contribute to phenotypic differences. For instance, the inhibitory effect of ZFP36 on growth of LN metastasis may not only be due to suppression of VEGF-C expression, but also to the pro-apoptotic effect of ZFP36 in tumor cells (first shown by Blackwell, PMID: 10763822). This pertains to the interpretation of Figure 6G-I, and to the discussion in general.

Response: We thank the reviewer for the comment. The interpretations of results and conclusions have been modified and discussed in the revised manuscript.

The manuscript stands in a line of studies demonstrating a tumor-suppressive effect of ZFP36, both in animal models (e.g. PMID: 12789264) and in human cancer (e.g. PMID: 21875902, PMID: 19491267, PMID: 19697322). In fact, some of these studies have already pointed to VEGF as an important target of ZFP36 with respect to its tumor-suppressive effects. To my knowledge, the first report of VEGF as a target of ZFP36 was in PMID: 17855506. The authors should present the background literature on ZFP36 to place the current study in proper perspective.

Response: We appreciate the comment. The introduction and appropriate references have been modified to better present the background on ZFP36.

page 2, second last line: decay of messenger RNA (mRNA)

Response: This has been corrected.

page 11, line 6: loss of ZFP36 expression

Response: This has been corrected.

Antibodies used for western blotting should be listed in the methods section.

Response: This point has been well taken. The antibodies used have been listed and provided in Supplementary Materials and Methods.

Reviewer #4 (Expertise: Esophageal cancer, Remarks to the Author):

In this study, the authors evaluated the significance of nicotine-mediated OTUD3 regulation and lymphangiogenesis in esophageal cancer. Interestingly, the OTUD3 expression was significantly associated with smoking status in clinical esophageal cancer patients, and the authors clarified the mechanism of OTUD3-regulated lymphangiogenesis via regulation of the ZFP36/VEGF-C axis. This report is intriguing; however, the current study does not meet the publishing criteria in this journal. Therefore, I raised several points to improve the content of the report.

Query

1. The authors focused on the nicotine effect against esophageal cancer cells, and the expression of OTUD3 in esophageal cancer cells was induced by nicotine treatment. Interestingly, the relation of OTUD3 expression in resected samples and smoking status was significant, and the data might be very interesting. However, esophageal cancer patients might stop smoking just before the resection surgery to prevent postoperative pulmonary complications. Why was the OTUD3 alteration in resected samples kept on an operative day? The comparison between current smokers and ever smoker may be important.

Response: We do appreciate the reviewer's comment. As stated by the reviewer, smoking patients were strongly demanded to stop smoking before the resection surgery to prevent postoperative pulmonary complications. Notably, the time it takes to clear the nicotine and its metabolites from the body depends on many factors, including age, sex, diet, type of tobacco used, and history of nicotine use. In particular, it takes people with long smoking history and older age much longer to eliminate the chemicals (<https://www.drugrehab.com/addiction/drugs/nicotine/nicotine-in-system/>). In the current study, we focused on the heavy-smoking esophageal cancer patients who had an overtime smoking history of no less than 20 pack-years ($PY \geq 20$) and were also older (median age: 58), suggesting that nicotine and its metabolites might not be quickly eliminated from the body, and would stay in the body before tumor resection.

To further address the reviewer's concern, we divided the heavy-smoking patients into the former ($n = 39$) and current categories ($n = 108$) as previously reported [1], in which former smokers were defined as those who quit smoking at least 1 year before surgery. Notably, the analysis revealed that OTUD3 expression was significantly correlated with smoking behavior (χ^2 test, $P < 0.001$, Cramer's $V = 0.381$), suggesting that OTUD3 expression was downregulated in the current smokers and could be recovered by quitting smoking (Supplementary Fig. 2b). These findings further indicate that OTUD3 expression is regulated by smoking in esophageal cancer.

Reference:

1. Kohei Shitara, et al. Heavy smoking history interacts with chemoradiotherapy for esophageal cancer prognosis: a retrospective study. *Cancer Sci.* 2010 Apr;101(4):1001-6.

How did the nicotine treatment regulate the PI3K/AKT/FOXO1 signaling for suppressing the OTUD3 expression in esophageal cancer?

Response: We do appreciate the comment and this point is important. Nicotine and its derivatives bind with the nicotinic acetylcholine receptors (nAChRs) to activate the PI3K/Akt signaling [1]. In particular, the alpha 7 nAChR is one of the most important nAChRs and is associated with human cancers [2]. Interestingly, our recent results showed that the alpha7 nAChR was increased in esophageal cancer cells (Supplementary Fig. 1g). Moreover, the alpha7 nAChR antagonist α -Bungarotoxin substantially impaired nicotine-induced AKT activation, FOXO1 inhibition, and OTUD3 downregulation in esophageal cancer cells (Supplementary Fig. 1h, i). These findings indicate that the alpha7 nAChR is essential for nicotine to regulate the PI3K/AKT/FOXO1/OTUD3 axis in esophageal cancer cells. These results have been incorporated into the revised manuscript.

Reference:

1. Grando SA. Connections of nicotine to cancer. *Nature reviews Cancer* 14, 419-429 (2014).
2. Schuller, H. M. Regulatory Role of the α 7nAChR in Cancer. *Curr. Drug Targets* 13, 680–687 (2012).

The relation of the OTUD3 expression and never-smoker, ever-smoker, and current smoker just before diagnosis should be shown as data.

Response: We do appreciate the reviewer's comment. To address the reviewer's concern, we further divided the heavy-smoking patients into the former (n = 39) and current categories (n = 108) as previously reported [1], in which former smokers were defined as those who quit smoking at least 1 year before surgery. Notably, a strong association [2] was found between OTUD3 expression and smoking behavior (χ^2 test, $P < 0.001$, Cramer's V = 0.381), suggesting that OTUD3 expression was downregulated in the current smokers and could be recovered by quitting smoking (Supplementary Fig. 2b). These findings further indicate that OTUD3 expression is regulated by smoking in esophageal cancer. These data have been incorporated into the revised manuscript.

Reference:

1. Kohei Shitara, et al. Heavy smoking history interacts with chemoradiotherapy for esophageal cancer prognosis: a retrospective study. *Cancer Sci.* 2010 Apr;101(4):1001-6.
2. Haldun Akoglu. User's guide to correlation coefficients. *Turk J Emerg Med.* 2018 Aug 7;18(3):91-93.

The prognostic value and expression significance of FOXO1, ZFP36, and VEGF-C in esophageal cancer patients with or without smoking history should be shown as data. Kaplan-Meier curves and clinicopathological tables should be used to show them. If the expressions were important for VEGF-C-related angiogenesis in esophageal patients with smoking history, the expression significance in non-smoker might be weak same as OTUD3.

Response: As recommended by the reviewer, the prognostic value and expression significance of FOXO1, ZFP36, and VEGF-C in esophageal cancer patients with or without smoking history have been analyzed. We found that low nuclear FOXO1 expression, low ZFP36 expression, or high

VEGF-C expression was significantly associated with heavy-smoking behavior in esophageal cancer patients (Supplementary Fig. 2f, 8b and Supplementary Table 3-5). Moreover, the Kaplan-Meier survival estimates showed that the signature of low nuclear FOXO1 expression, low ZFP36 expression, or high VEGF-C expression performed better in predicting RFS and OS in heavy-smoking patients than in the non-smokers (Supplementary Fig. 2g-h and Supplementary Fig. 8c-d). These results have been incorporated into the revised manuscript.

In this study, the authors clarified the interesting relationship between nicotine, PI3K/AKT/FOXO1, OTUD3, ZFP36, and VEGFC in esophageal cancer. However, the story might be complex. Moreover, the relation should be validated in clinical esophageal cancer samples statistically.

Response: We do appreciate the comment. In this study, via RNA-seq screening we identified OTUD3 as a potential downstream candidate of nicotine. We then clarified that nicotine reduced OTUD3 expression via PI3K/AKT/FOXO1 signaling. Notably, nuclear FOXO1 expression significantly correlated with OTUD3 expression in patient specimens (χ^2 test, $P < 0.001$, Cramer's $V = 0.779$; Supplementary Fig. 2e and Supplementary Table 3), suggesting that this regulation was clinically relevant. Furthermore, we found that OTUD3 induced VEGF-C mRNA decay by stabilizing ZFP36, which also showed significant correlations in patient specimens (Fig. 7a-c and Supplementary Table 2). Importantly, patients with combined low OTUD3 and high VEGF-C expression suffered the worse RFS and OS, especially in the heavy-smoking patients (Fig. 7d-f), further suggesting that the OTUD3/VEGF-C axis was indeed associated with poor clinical outcomes of esophageal cancer. The relations between the expression levels of these genes have been provided in the revised manuscript.

IRB-approved number should be described in the materials and methods section.

Response: The IRB-approved number (GZR2016-111) has been included in the METHODS section.

Antibody information of p63 for IHC and GAPDH and tubulin for WB should be described in the materials and method section. Why did the authors change the antibodies of OTUD3 for IHC and WB

Response: As recommended, antibody information of p63 for IHC and GAPDH and tubulin for WB has been provided in METHODS and Supplementary Material and Methods. We used two different OTUD3 antibodies for IHC and WB, as they are tested suitable for IHC and WB, respectively. The specificities of these two antibodies were validated in esophageal cancer cells and mice tumors with OTUD3 overexpression or depletion (Fig. 3h and Supplementary Fig. 3d, e).

The information on adjuvant therapy in esophageal cancer patients should be described in the materials and method section.

Response: Adjuvant therapy was normally conducted in esophageal cancer patients after tumor

resections, with only some early-stage tumors did not receive it. Adjuvant therapy had no impact on gene expression in the patient specimens used in this study. The information on adjuvant therapy has been described in METHODS and the number was included in Supplementary Table 1.

Reviewer #5 (Expertise: DUBs, cancer, Remarks to the Author):

The authors investigated the link between nicotine addiction and esophageal tumorigenesis, notably lymphatic metastasis of esophageal cancer cells. The authors found that the deubiquitinase OTU domain-containing protein 3 (OTUD3) is downregulated by nicotine administration and its expression is correlated with poor prognosis in heavy-smoking esophageal cancer patients. The results support a model whereby nicotine-mediated OTUD3 downregulation is associated with tumor-induced lymphangiogenesis. Mechanistically, they found that OTUD3 directly interacts with ZFP36 ring finger protein (ZFP36) resulting in its stabilization. This effect is associated with an inhibition of FBXW7-mediated K48-linked polyubiquitination. Next, the authors established that ZFP36 binds the mRNA of VEGF-C at its 3'-UTR and to promote its degradation. The authors proposed that nicotine induces downregulation of OTUD3 which leads to ZFP36 downregulation, which in turn promotes VEGF-C production and lymphatic metastasis in esophageal cancer.

Overall, this is a very interesting study that links nicotine addiction to OTUD3 and VEGF-C signaling and their impact on lymphatic metastasis. The OTUD3-VEGF-C signaling might be potentially exploited for the treatment of human esophageal cancer. The study is quite novel, and the data are in general convincing. The statistics appear adequate. I think this study could be of interest to a wide audience of researchers and clinicians. Nonetheless, I think the following comments need to be addressed before consideration of the manuscript for publication.

1) In Figure 1B, immunoblotting for OTUD3 protein in the various cell types treated with cigarette smoke cigarette need to be shown to corroborate the mRNA expression data.

Response: As recommended by the reviewer, the immunoblotting for OTUD3 protein was incorporated in Fig. 1b, showing that CSE downregulated OTUD3 proteins in esophageal cancer cell lines.

2) Figure 2, The authors need to be explicit in their figures and legend to indicate whether they conducted overexpression or RNAi knockdown.

Response: Appropriate modifications have been made to figure legends.

3) In general, the authors need to better describe the figures in the legends. In addition, the authors need to indicate the number of repetitions for all figures, e.g., western blots and RNA pull downs.

Response: As recommended, appropriate modifications have been made to figure legends.

4) Figure 4. As described in material and methods, the IP-immunoblotting was not conducted under denaturing conditions. The authors could not exclude the possibility that the ubiquitination signals observed by immunoblotting might be contributed by additional ZFP36-associated proteins. Thus, the

authors should repeat some of these immunoprecipitation experiments by first denaturing the extract in 1% SDS in conjunction with heat inactivation and then dilute the samples in an adequate buffer before conducting the IP. This will help excluding the possibility that other proteins are ubiquitinated.

Response: We thank the reviewer for this comment and this point is important. We further conducted some of the ubiquitination assays using the denaturing method. As shown in Supplementary Figure 5d-e, IP assays with the denaturing conditions showed that nicotine promoted ZFP36 ubiquitination, while ectopic expression of OTUD3, but not OTUD3^{C76A}, repressed it. These results further indicated that OTUD3 inhibited the polyubiquitination of ZFP36.

5) The authors have the tendency to overstate the conclusions of the manuscript. For instance, the authors state “Nicotine-mediated OTUD3 downregulation promoted lymphatic metastasis by inducing substantial tumor-induced lymphangiogenesis”.

But the authors did not carefully demonstrate that modulation of OTUD3, induces tumor-induced lymphangiogenesis. The results shown by the authors in figure 2 are quite preliminary.

Response: We are sorry for the improper interpretations. In Figure 2, we found that nicotine promoted lymphangiogenesis and lymph node metastasis, and these effects were abrogated by OTUD3 overexpression. Notably, the conditioned medium from OTUD3-silencing esophageal cancer cells enhanced the migration and tube formation of hLECs (Supplementary Fig. 3e-g), suggesting that downregulation of OTUD3 plays a role in tumor-induced lymphangiogenesis. Interpretations of the results and the conclusion have been revised to avoid overstatements.

6) The authors need to repeat some of the key experiments on the biological impact of OTUD3 with a catalytic dead mutant to further corroborate findings on substrate deubiquitination.

Response: We do appreciate the comment and this point is important. We further examined whether OTUD3 with a catalytic dead mutant lost its role in regulating VEGF-C expression and tumor-induced lymphangiogenesis. As expected, OTUD3^{C76A} did not change VEGF-C mRNA expression (Supplementary Fig. 4g). Moreover, conditioned medium from OTUD3^{C76A} overexpressing esophageal cancer cells had no significant effects on the VEGFR signaling, migration, and tube formation of hLECs (Supplementary Fig. 4h-j). These results reveal that the catalytic activity of OTUD3 is indispensable for its role in tumor-induced lymphangiogenesis.

7) The authors consider the E3 ligase FBXW7 for targeting the ubiquitination of ZFP36. The authors could consider conducting shRNA knockdown of this ubiquitin ligase and determine the impact on ZFP36 stability and VEGF-C signaling. For example, does FBXW7 depletion rescues the effects of OTUD3 depletion. In addition, FBXW7 is known as a tumor suppressor, and it should be discussed how this tumor suppressor activity could be integrated within the OTUD3-ZEP36-VEGF-C signaling axis.

Response: We thank the reviewer for this comment and these suggestions have been well taken. As shown in Supplementary Fig. 5g-i, downregulation of FBXW7 inhibited polyubiquitination of ZFP36, increased ZFP36 protein expression, and reduced VEGF-C expression in OTUD3-silencing esophageal cancer cells. These findings indicate that downregulation of FBXW7 rescues the effects of OTUD3 depletion on ZFP36 and VEGF-C expression.

The dynamic of polyubiquitination is balanced both by E3 ligase-induced ubiquitination and DUBs-mediated deubiquitination. Interestingly, our recent findings indicated that the FBXW7 E3 ligase was not affected by cigarette smoke extract (CSE) or nicotine in esophageal cancer cells (Fig. 1a and Supplementary Fig. 5j). Thus, these findings suggest that nicotine decreases ZFP36 expression by downregulating OTUD3, but not through alteration of FBXW7, leading to VEGF-C elevation. These results have been incorporated and discussed accordingly

8) It might be interesting to determine by confocal microscopy, the subcellular localization of OTUD3 and ZFP36 by immunostaining to determine whether these factors have similar subcellular compartmentalization.

Response: We thank the reviewer for this comment. In fact, we had used the proximity ligation assay (PLA, Sigma-Aldrich) to examine the endogenous protein interaction between OTUD3 and ZFP36. The PLA assay generates fluorescent spot signals using oligonucleotide labeled secondary antibodies (PLA probes), once two primary antibodies were in close proximity. These PLA signals can be quantified (counted) and assigned to a specific subcellular location. As shown in Fig. 4c, we found a cytoplasmic association of OTUD3 and ZFP36 in KYSE180, and this interaction was significantly decreased by nicotine treatment.

To further address the reviewer's concern, dual-IF staining of OTUD3 and ZFP36 was performed in KYSE180 and KYSE520 cells. Consistently, the staining showed cytoplasmic colocalization of OTUD3 and ZFP36 in esophageal cancer cells (Supplementary Fig. 5b). These results have been incorporated into the revised manuscript.

9) The mass spectrometry data obtained through the purification of OTUD3-associated proteins should identify several peptides for each protein. The authors need to provide the data for all peptides counts and deposited in a resource database.

Response: As recommended, the mass spectrometry data have been deposited in the iProX database, with the reference number #PXD028751. This information has been incorporated.

10) The authors show the effects of knockdowns, sometimes with two shRNAs, sometimes with one shRNA only. It should be two shRNAs, ideally throughout the manuscript. This will ensure that the observed effects are not due to off-target effects. In addition, the authors indicate shRNA and siRNA, this has to be clarified. If siRNAs are used, two different siRNAs should be used to exclude potential

off target effects. In the shRNA experiments, what is control? Non-transfected, empty vector, or a scrambled sequence? This also needs to be indicated.

Response: As recommended by the reviewer, experiments have been performed with two shRNAs or siRNAs throughout the manuscript. These results further strengthen the conclusion of the study and have been incorporated.

In shRNA experiments, the scrambled sequence is used as the control. The figure labels and legends have been revised to make it clear.

11) The discussion should be broadened to put everything in context and to cover the known effect and mechanisms of action of nicotine.

Response: This suggestion has been well taken. The roles and mechanisms of nicotine in cancer have been discussed in the revised manuscript.

Other comments

12) List the source and catalogues numbers of all antibodies used, e.g., anti-p-Akt, anti-Akt, anti-p-ERK1/2, anti- ERK1/2, anti-p-p38, anti-p38 antibodies, Ubiquitin antibody, etc.

Response: As recommended, the information of antibodies used has been provided in the Supplementary Materials and Methods.

13) The histograms need to be uniform. Sometimes histograms are shown with the error bar only (Ex: Fig 1), but all points are shown with the error bar in figure 6G, I. Include error bar with individual points in all histograms.

Response: This point has been done. Appropriate modifications have been made to histograms.

14) Include scale barre in all IHC pictures (at least one picture per panel).

Response: As recommended, scale bars have been added to the IHC pictures.

15) In the introduction, the authors state Deubiquiyinase are pleiotropic players... This might need to be reformulated

Response: As suggested, the text has been modified to “Deubiquitinase (DUBs) can exert pleiotropic functions by trimming or removing ubiquitin chains from diverse substrate proteins”.

16) Correct all typos, eg., Ployubiquitination analysis.

Response: The manuscript has been checked thoroughly and typos have been corrected.

17) Perhaps consider changing the tense of some sentences. In material and methods e.g., Details of the IHC method are (instated of were) provided in the supplementary information.

In the abstract, Here we show (instead of Here we showed). At the mechanistic level, OTUD3 directly interacts (instead of interacted) with ZFP36 ring finger protein (ZFP36) and stabilized it by inhibiting FBXW7-mediated K48-linked polyubiquitination.

Response: The suggestion has been well taken. Appropriate modifications have been made.

18) Manuscript to format according to the style of Nature Comm.

Response: The format of the manuscript has been modified according to the style of Nature Comm.

REVIEWERS' COMMENTS

Reviewer #1 (Remarks to the Author):

The authors have addressed all points raised. I have only one minor comment: please correct the references for LYVE-1 and podoplanin antibodies as markers for lymphatics.

Tadayon and Song references are not correct here. Authors should cite the original source that demonstrated specificity of these antibodies for lymphatics; for LYVE-1 work by David Jackson, and for podoplanin, work by Dontcho Kerjaschki.

Reviewer #2 (Remarks to the Author):

In efforts to better understand cigarette-smoking induced lymphatic metastasis in human esophageal cancer, the authors investigate a specific deubiquitinase, OTUD3, which is modified by nicotine exposure, promoting lymphatic metastasis in-vivo. The authors exemplify noteworthy results which are original and significant to the field. As a reviewer, my most serious concerns were about transparency of methods. I find the authors responses and revisions to the raised concerns to be adequate. I appreciate the authors time and attention to detail for all concerns raised.

Reviewer #3 (Remarks to the Author):

My concerns have been addressed in the revised manuscript. I am looking forward to seeing the story published.

Reviewer #4 (Remarks to the Author):

This study data is interesting; however, several limitations considerably reduce the significance of this study.

Reviewer #5 (Remarks to the Author):

The authors have addressed most of my comments

I have some additional comments:

1) In the abstract:

-ZFP36 binds with the VEGF-C 3'-UTR and recruits the RNA degrading complex to induce (use induce instead of conduct) its rapid mRNA decay.

-Sentence with thus (remove Thus) and directly start: Downregulation of OTUD3 and ZFP36 is essential for nicotine-induced VEGF-C production and lymphatic metastasis in esophageal cancer.

-Include ZFP36 in the keywords

2) In Figures:

-The authors state that they conducting IP under denaturing conditions, but this is not stated in the Polyubiquitination analysis in Methods.

-In many occasion in figure legends, remain unclear the number of replicates (technical or biological ?) (e.g., Figure 3a, b,c many panels of Figure 4 and Figure 5) The authors state at the end of figure 4, Immunoblot is representative of three independent experiments. Is this for all panels or panel 4j only ? Please carefully revise all figures.

-Magnification bars are missing in some panels.

-The mass spectrometry data should be also provided as a table to show peptides for each protein (and counts).

Response to reviewers' comments and suggestions:

Reviewers' comments:

Reviewer #1 (Remarks to the Author):

The authors have addressed all points raised. I have only one minor comment: please correct the references for LYVE-1 and podoplanin antibodies as markers for lymphatics.

Tadayon and Song references are not correct here. Authors should cite the original source that demonstrated specificity of these antibodies for lymphatics; for LYVE-1 work by David Jackson, and for podoplanin, work by Dontcho Kerjaschki.

Response: We thank the reviewer for the positive comment. As requested, the references have been corrected in the revised manuscript.

Reviewer #2 (Remarks to the Author):

In efforts to better understand cigarette-smoking induced lymphatic metastasis in human esophageal cancer, the authors investigate a specific deubiquitinase, OTUD3, which is modified by nicotine exposure, promoting lymphatic metastasis in-vivo. The authors exemplify noteworthy results which are original and significant to the field. As a reviewer, my most serious concerns were about transparency of methods. I find the authors responses and revisions to the raised concerns to be adequate. I appreciate the authors time and attention to detail for all concerns raised.

Response: We thank the reviewer for his (her) appreciation of our efforts in addressing the concerns.

Reviewer #3 (Remarks to the Author):

My concerns have been addressed in the revised manuscript. I am looking forward to seeing the story published.

Response: We thank the reviewer for his (her) appreciation of our efforts in addressing the concerns.

Reviewer #4 (Remarks to the Author):

This study data is interesting; however, several limitations considerably reduce the significance of this study.

Response: We thank the reviewer for his (her) appreciation of our efforts in addressing the concerns.

Reviewer #5 (Remarks to the Author):

The authors have addressed most of my comments

I have some additional comments:

1) In the abstract:

-ZFP36 binds with the VEGF-C 3'-UTR and recruits the RNA degrading complex to induce (use induce instead of conduct) its rapid mRNA decay.

-Sentence with thus (remove Thus) and directly start: Downregulation of OTUD3 and ZFP36 is essential for nicotine-induced VEGF-C production and lymphatic metastasis in esophageal cancer.

-Include ZFP36 in the keywords

Response: We thank the reviewer for the positive comment. As suggested, appropriate modifications have been made to the abstract.

2) In Figures:

-The authors state that they conducting IP under denaturing conditions, but this is not stated in the Polyubiquitination analysis in Methods.

-In many occasion in figure legends, remain unclear the number of replicates (technical or biological ?) (e.g., Figure 3a, b,c many panels of Figure 4 and Figure 5) The authors state at the end of figure 4, Immunoblot is representative of three independent experiments. Is this for all panels or panel 4j only ? Please carefully revise all figures.

-Magnification bars are missing in some panels.

-The mass spectrometry data should be also provided as a table to show peptides for each protein (and counts).

Response: We thank the reviewer for the comments and these points have been well taken. As requested, appropriate modifications have been made to the methods, figure legends, magnification bars, and mass spectrometry data in the revised manuscript.